# The Interactions between Polyphenols and Microorganisms, Especially Gut Microbiota

**DOI:** 10.3390/antiox10020188

**Published:** 2021-01-28

**Authors:** Małgorzata Makarewicz, Iwona Drożdż, Tomasz Tarko, Aleksandra Duda-Chodak

**Affiliations:** Department of Fermentation Technology and Microbiology, Faculty of Food Technology, University of Agriculture in Krakow, 30-149 Kraków, Poland; malgorzata.makarewicz@urk.edu.pl (M.M.); iwona.drozdz@urk.edu.pl (I.D.); tomasz.tarko@urk.edu.pl (T.T.)

**Keywords:** intestinal microbiota, inhibition, metabolism, catabolism, biotransformation, bioactive compounds, health, metabolites, diversity

## Abstract

This review presents the comprehensive knowledge about the bidirectional relationship between polyphenols and the gut microbiome. The first part is related to polyphenols’ impacts on various microorganisms, especially bacteria, and their influence on intestinal pathogens. The research data on the mechanisms of polyphenol action were collected together and organized. The impact of various polyphenols groups on intestinal bacteria both on the whole “microbiota” and on particular species, including probiotics, are presented. Moreover, the impact of polyphenols present in food (bound to the matrix) was compared with the purified polyphenols (such as in dietary supplements) as well as polyphenols in the form of derivatives (such as glycosides) with those in the form of aglycones. The second part of the paper discusses in detail the mechanisms (pathways) and the role of bacterial biotransformation of the most important groups of polyphenols, including the production of bioactive metabolites with a significant impact on the human organism (both positive and negative).

## 1. Introduction

The intestinal microbiome plays an important, if not crucial, role in the metabolism of chemical compounds delivered with food, especially those that are undigested in the upper digestive tract. The enormous number of bacterial cells inhabiting the large intestine forms a complex ecosystem called the “intestinal microbiome”. The word microbiome was introduced for the first time in 2001 to define the collective genomes of the microbiota [1]. Since then, much research and many projects were dedicated to assessing the impact of intestinal microbiota on a host’s health, especially by determining its role in food metabolism, xenobiotics biotransformation and various disease development.

It is estimated that the microbiota of an adult is composed of ~10^14^ bacteria cells [2] belonging to 1000–1150 species, with each individual harboring at least 160 species (usually about 500 species) [3]. Based on the large-scale 16S rRNA or metagenomic studies, scientists stated that ~80% of the bacteria identified by molecular tools in the human gut are uncultured and hence can be characterized only by metagenomic studies [4,5]. There are significant interindividual differences in the bacterial species found in the gastrointestinal tract. The composition, as well as the ratio of different species that form the intestinal microbiome, is very diverse within the human population, and each individual has his or her own unique profile of microbial species, which can be compared to a fingerprint. The differentiation of gut microbiota composition and profile is caused by the influence of multiple and diverse factors, such as age, origin, geographical location, environment, dietary habits (including probiotics), health, the application of antibiotics or even in the way an individual is born [6,7,8]. However, despite the great diversity of bacterial species, the majority of them belong to only four bacterial phyla: Firmicutes (64%), Bacteroidetes (23%), Proteobacteria (8%) and Actinobacteria (3%), whereas other taxons highly diverse [2]. Among the key functions of microbiota is occupying the intestinal surfaces and production of antimicrobial compounds that prevent the invasion of pathogens. Both commensal bacteria and gut pathogens (such as *Salmonella*, *Shigella*, *Helicobacter*, *Vibrio*, *Campylobacter*, *Yersinia*, *Clostridia*, *Aeromonas*, *Listeria*, *Streptococcus*, and *Staphylococcus*, as well as pathogenic strains of *Escherichia coli*, *Klebsiella pneumoniae*, *Enterococcus faecalis*) require similar ecological niches to colonize and proliferate in the intestine [9,10]. Therefore, various mechanisms designed to compete with each other have evolved. Commensal bacteria produce bacteriocins that specifically inhibit members of the same or similar bacterial species (e.g., *E. coli* versus pathogen enterohemorrhagic *E. coli*). Commensal bacteria produce short-chain fatty acids and cause pH reduction, thereby preventing the colonization by pathogens whose optimal pH for growth is neutral [11]. An altered bacterial community structure may facilitate the gut colonization by enteric pathogens but can also favor the overgrowth of potentially harmful subsets of indigenous bacteria, like virulent *E. coli* or *Clostridium difficile*.

The great diversity of bacterial species forming the gut microbiota implicates the large number of genes which they contain [2] and the enormous metabolic capacity of the intestinal microbiome, which is approximately 100-fold greater than that of the human liver [2,12]. The intestinal microbiota is equipped with a large set of different enzymes able to hydrolyze glycosides, glucuronides, sulfates, amides, esters and lactones through the action of enzymes such as α-rhamnosidase, β-glucuronidase, β-glucosidase, sulfatase and various esterases. Other reactions catalyzed by the gut microbial enzymes are aromatic ring cleavage, reductions (reductases, hydrogenases), decarboxylation (decarboxylase), demethylation (demethylase), isomerization (isomerase), and dehydroxylation (dehydroxylase) [13,14].

Intestinal bacteria contribute to the breakdown of polysaccharides and polyphenols as well as participate in the synthesis of vitamins (K, B12) and amino acids [8]. Many metabolites produced by gut microbiota are involved in various important physiological processes of the host, including energy metabolism and immunity. For example, the essential aromatic amino acid tryptophan can be metabolized by—among others—*Peptostreptococcus russellii*, *Clostridium sporogenes*, and *Lactobacillus* spp. to indole derivatives, which are ligands for aryl hydrocarbon receptor (AhR). This transcription factor plays an important role in the human immunological response, and via modulating T cell differentiation, Th17 development and IL-22 production may inhibit inflammation [15]. Branched-chain amino acids (BCAAs) (such as leucine, isoleucine, and valine) are essential amino acids that possess an aliphatic side chain with a branch, and that cannot be synthesized by humans. Therefore, they are provided by diet or synthesized by gut microbiota. The main species that contribute to the BCAAs production are *Prevotella copri* and *Bacteroides vulgatus* [15].

To date, thousands of microbial metabolites with known and unknown functions have been identified as components of the human metabolome. Well-known are short-chain fatty acids (SCFAs) with the acetate, propionate, and butyrate, being the metabolite of resistant starch and dietary fiber fermentation [15]. SCFAs are generally considered to have beneficial effects on host health, and they modulate metabolism, inflammation, hormone production, lipogenesis, and gut homeostasis. Gut microbiota can also metabolize dietary L-carnitine, choline, and lecithin into trimethylamine (TMA), which is then converted to trimethylamine-N-oxide (TMAO) in the liver of a host.

Owing to the multitude of direct and indirect interactions with the host organism, the intestinal microbiome is hence closely linked to the health of a host [16,17]. Dysbiosis, an imbalanced or disturbed microbiota composition, may play a significant role in etiology or the development of various gastrointestinal diseases such as inflammatory bowel disease (IBD), irritable bowel syndrome (IBS), colon cancer, and antibiotic-associated diarrhea [12,18]. The gut microbiota also plays a critical role in the transformation of dietary polyphenols into absorbable biologically active compounds. It is estimated that about 90–95% of the total polyphenol intake remains unabsorbed and colonic bacteria act enzymatically on their backbone, producing metabolites with a different physiological significance [19].

## 2. The Structure and Role of Polyphenols

Polyphenols are secondary metabolites playing an important role in plant tissues. They provide the color to flowers and fruits (mainly anthocyanins), which attracts pollinators and seed dispersers; they are also responsible for flavor in fruit and vegetables, as well as protecting plant tissues against herbivores and other biotic and abiotic stressors, like UV radiation, cold, heat or salinity [20]. Flavonoids take part in energy transfers, the regulation of photosynthesis and morphogenesis, regulation of growth factors, and sex determination and—due to antimicrobial activity—protect against the spread of pathogens in plant tissues [21]. Polyphenols also influence human health. Because of antioxidant properties and free-radical scavenging activity, they are believed to protect against various diseases, e.g., cancer, stroke and myocardial infarction, cardiovascular diseases, and some immunological and neurological disorders; they are also thought to have a beneficial impact on humans with diabetes and obesity [22,23,24,25,26,27,28,29,30,31]. Several in vitro and in vivo animal studies have demonstrated the antioxidant and anti-inflammatory effects of polyphenols in the brain–liver–gut axis [32], and polyphenols have been shown to target different stages of the inflammatory cascade to reduce the severity of inflammation. Polyphenols can also modulate various signal pathways, for example, through interaction with AMP-activated protein kinase (AMPK), CCAAT/enhancer-binding protein α (C/EBPα), peroxisome proliferator-activated receptor γ (PPARγ), and peroxisome proliferator-activated receptor-gamma coactivator 1-alpha (PGC-1α), sirtuin 1, and sterol regulatory element-binding protein-1c (SREBP1c) involved mainly in cellular energy metabolism and adipogenesis, as well as uncoupling proteins 1 and 2 (UCP1 and UCP2), and NF-κB that regulate antioxidant and anti-inflammatory responses [28].

Polyphenols are a large group of compounds that comprise phenolic acids, flavonoids, tannins, lignans, stilbens and coumarins (Figure 1 presents the chemical structure of flavonoids, Figure 2—non-flavonoids). In a human diet, they are provided mainly by plant food such as fruit, vegetables, tea, wine, coffee, and cocoa. However, even when phenolic compounds occur in the human diet in large quantities, they do not always show high biological activity after consumption.

Polyphenols’ influence on human health depends both on their amount of food and on their bioavailability, bioaccessibility and the biological activity of metabolites produced in the human body. Some phenolic compounds have limited absorption in the digestive tract, while the others undergo an intensive metabolism to derivatives with a lower activity or they undergo a rapid elimination (degradation). The uniqueness of the gut microbiota composition causes that in one individual, a given polyphenol will undergo bacterial metabolism and will have an effect (beneficial or negative), whereas in another human being, the metabolism of the same polyphenol will follow a different path, and there will be no effect.

## 3. The Impact of Polyphenols on Microorganisms and the Mechanism of Their Action

Herbs and spices have a long history of being used as natural food preservatives and within folk medicine. It is due to substances with the antimicrobial activity they contain, such as flavonoids, anthocyanins, alkaloids, glycosides, saponins, coumarins, tannins, vitamins, phenolic acids and many more. Essential oils (a complex mixture of various bioactive compounds, inter alia, polyphenols), plant extracts and pure polyphenols are large groups of compounds with strong antibacterial properties. It has been proven many times that thyme, oregano, rosemary, sage, mint and other herbs and spices can inhibit Gram-positive and Gram-negative bacteria, including pathogens [20]. There are many scientific reports and publications demonstrating that pure polyphenols or bioactive compounds present in various types of plant preparations (e.g., aqueous, ethanolic or methanolic extracts, essential oils, enriched extracts) can exert both negative and positive impact on microorganisms. Some examples of such impact are presented in Table 1. 

As can be seen in the table, not only pathogenic bacteria growth can be inhibited by polyphenols. It was proved that some beneficial microorganisms, inter alia lactic acid bacteria and probiotics strains, can also be inhibited. However, the bacteriostatic or bactericidal effect depends both on the polyphenol structure and bacteria species. Susceptibility to some polyphenols was also proved to be strain-dependent [43,47,50,64]. It was proved that some polyphenols exert a beneficial influence on bacteria. They can stimulate growth or at least change the composition of the microbiome in favor of beneficial bacteria such as *Bifidobacterium* and *Lactobacillus*, which both contribute to gut barrier protection; *Akkermansia muciniphila* and *Faecalibacterium prausnitzii* that possess anti-inflammatory effect by inhibiting the activation of NF-kB; and *Roseburia* sp.—the butyrate-producer [15]. *Akkermansia muciniphila* is an anaerobic, mucin-degrading bacterium residing in the healthy intestinal tract of a host that is believed to have several health benefits in humans [89]. Many studies show that various diseases, e.g., obesity, type II diabetes and inflammatory bowel diseases have an association with reducing *A. muciniphila* abundance [90]. Taking into account that polyphenols have been proven to increase the growth of *Akkermansia muciniphila* [72,73,75,76,77], the beneficial impact of polyphenols on human health may result from other than antioxidant activity. All these findings (Table 1) suggest that polyphenols reaching the large intestine may not only be catabolized to small phenolic acids but also elicit potentially beneficial effects of intestinal probiotic bacteria. Taken together, polyphenols appear to be able to alter gut microecology and, by affecting the total number of beneficial species in the gut, may confer positive gut health benefits.

## 4. Mechanism of Antibacterial Activity of Polyphenols

As was reported above, the influence of pure polyphenols and plant extract, as well as the strength of that impact on bacteria, differs depending on the kind of both phenolic compounds and bacteria strain. The mechanism of antimicrobial activity of polyphenols against bacteria can differ and also depends both on the polyphenol type and bacteria species. Among the most important mechanisms of the antibacterial action of polyphenols are [58,91,92,93,94,95,96,97,98]:Reactions with proteins;Inhibition of nucleic acid synthesis by bacterial cells or DNA damage;Interaction with the bacterial cell wall or inhibition of cell wall formation;Alteration of cytoplasmic membrane function, such as modifications of the membrane permeability or fluidity, cytoplasmic membrane damage and—in the result—the membrane disruption;Inhibition of energy metabolism;Changes in cell attachment and inhibition of biofilm formation;Substrate and metal deprivation.

### 4.1. Reactions with Proteins

The antibacterial activity of flavonoids may result from their ability to form complexes with proteins through nonspecific forces such as hydrogen bonding and hydrophobic effects, as well as by covalent bond formation [99]. Due to protein binding by polyphenols, they are sequestered into soluble or insoluble complexes, which affects the function of both polyphenol and protein [100]. Proteins modified by polyphenols binding have some amino acids blocked or undergo conformation transitions, which can cause changes in protein structure, solubility, hydrophobicity, thermal stability, and the isoelectric point. In consequence, protein–phenolic complexation leads to changes in their physicochemical and biological properties, including the digestibility and utilization of food proteins as well as the activity of digestive enzymes [101]. It has been demonstrated that naturally occurring polyphenols, e.g., condensed tannins, can inhibit a number of digestive enzymes, including α-glycosidase, α-amylase, lipase, pepsin, trypsin, and chymotrypsin, changing the availability of nutrients and hence modulating the microbiota composition [102,103,104,105,106]. Furthermore, polyphenols can bind to important bacterial proteins such as adhesins, enzymes, cell envelope transport proteins and—by inactivating them—exert an antimicrobial impact. On the other hand, polyphenols complexation with proteins may influence the bioaccessibility and activity of phenolic compounds.

Quinones are known to complex irreversibly with nucleophilic amino acids in proteins, which leads to inactivation and loss of function in the proteins. They possibly interact with cell wall polypeptides, membrane-bound enzymes and surface-exposed adhesins of pathogenic bacteria [107].

Kaempferol-3-rutinoside (nicotiflorin) **(130)** was demonstrated to inhibit *Streptococcus mutans*. The affected protein was sortase A, a membrane enzyme that actively plays a crucial role in bacteria adhesion and the invasion of host cells [108]. A purified *S. mutans* sortase A was inhibited by curcumin **(216)** at a half-maximal inhibitory concentration; curcumin was also found to release the Pac protein to the supernatant and reduce *S. mutans* biofilm formation [109]. The sortase enzymes (cysteine transpeptidases) are used by Gram-positive bacteria to display proteins in a cell surface (e.g., glycoproteins), and they can attach to proteins in the cross-bridge peptide of the cell wall. Hence they are an important virulence factor. Morin **(110)**, myricetin **(114)**, and quercetin **(111)** exhibited strong inhibitory activity against sortase A and B from *S. aureus* [110].

Some flavones were active against *Escherichia coli* by forming complexes with extracellular and soluble proteins [111]. Chen et al. [112] proved that baicalein **(45)** decreased the expression of intracellular adhesin in *S. aureus*. EGCG can bind to porins, so probably this way, it affects the permeability of the outer membrane of Gram-negative bacteria via porin pores [113].

Nakayama et al. [114] demonstrated with two-dimensional electrophoresis that epigallocatechin gallate **(18)** strongly interacted with one of the outer membrane porin proteins of *E. coli*, especially with basic amino acids such as Arg, Lys and His. The docking simulation revealed that EGCG enters into the porin pore and binds to Arg residues present on the inner surface of the pore channel through hydrogen bonding, resulting in inhibition of the porin function.

It also has been shown that tea catechins [115] and various polyphenols [20] have the capacity to sensitize strains of methicillin-resistant *Staphylococcus aureus* to antibiotics. Taylor et al. [115] postulated that catechin gallates (compounds **13**–**18**) intercalate into phospholipid bilayers, and probably they affect both virulence and antibiotic resistance by perturbing the function of the key processes associated with the bacterial cytoplasmic membrane.

The inhibitory impact of polyphenols on the action of the bacterial efflux pump, which changes transport through the cell wall and cytoplasmic membrane, is also taken into account [91,116]. It has been demonstrated that quinones and chalcones are substrates of bacterial efflux pumps and could be used in combination with the efflux inhibitors in order to improve the accumulation of the drug in the cells to fight against MRSA infections [117]. Kaempferol **(108)** and galangin **(109)** were found to be effective efflux pump inhibitors in *S. aureus* [118].

Flavonoids can modulate the activity of bacterial enzymes, which are crucial for cell life, such are those catalyzing the synthesis of cell wall elements, cell membrane fatty acids or ATP. Fatty acid synthase II (FAS-II) is a key enzyme for the synthesis of fatty acids building the bacterial membranes. It catalyzes fatty acid chain elongation, from 16–24 carbons obtained de novo by FAS-I to long-chain fatty acids of 36–48 carbons as well as mycolic acids [92]. Flavonoids such as isoliquiritigenin **(142)**, butein **(144)**, fisetin **(107)** and 2,2′,4′-trihydroxychalcone **(139)** inhibited FAS-II, thus preventing the growth of *Mycobacterium smegmatis* [119].

Epigallocatechin gallate (EGCG) **(18)** and the related tea catechins potently inhibited both the FabG and FabI reductase steps in the fatty acid elongation cycle [120]. The authors suggested that the presence of the galloyl moiety was essential for inhibitory activity, and EGCG was a competitive inhibitor of FabI and a mixed-type inhibitor of FabG, demonstrating that EGCG interfered with cofactor binding in both enzymes. Furthermore, EGCG inhibited acetate incorporation into fatty acids in vivo. Molecular docking studies conducted by Xiao et al. [121] revealed the importance of the 3-*O*-galloyl or 3-*O*-glycosides side chain at the flavonoid pyran ring in the mechanism of the inhibition of reductase flavoprotein, dihydroorotate dehydrogenase (PyrD), dihydrofolate reductase (DYR), NADH-dependent enoyl-ACP reductase, and the DNA gyrase subunit in *E. coli*. Results obtained in the study demonstrated that EGCG has the strongest binding with NADH-dependent enoyl-ACP reductase (FabI) in comparison with other flavonoids, while quercitrin **(111)** was also the strongest inhibitor of DNA gyrase subunit B (GyrB) among tested 19 flavonoids. The results also indicated that flavonoids that have galloyl moieties, such as EGCG **(18)**, (−)-catechin gallate **(16)**, (−)-epicatechin gallate **(17)**, and (−)-gallocatechin gallate **(13)**, exhibited higher binding affinities to PyrD, FabI, and DYR than their cognates lacking the galloyl group, i.e., (−)-epigallocatechin **(14)**, (+)-catechin **(7)**, (−)-epicatechin **(12)**, and (−)-gallocatechin **(11)**, respectively.

Quercetin **(111)**, apigenin **(38)**, and sakuranetin **(84)** inhibited the activity of β-hydroxyacyl-acyl carrier protein dehydratase from *Helicobacter pylori* (HpFabZ), which is necessary for bacterial fatty acid biosynthesis. These three flavonoids are all competitive inhibitors against HpFabZ by binding to the substrate tunnel and preventing the substrate from accessing the active site [122]. Similarly, β-ketoacyl acyl carrier protein synthase (KAS) III is a key catalyst in bacterial fatty acid biosynthesis. The docking studies between *Enterococcus faecalis* KAS III (efKAS III), and flavonoids proved that naringenin **(81)**, eriodictyol **(94)**, and taxifolin **(134)**, with high-scoring functions and good binding affinities, docked well with efKAS III, causing the *E. faecalis* growth inhibition. Hydrogen bonds between the 5- and 4′-hydroxy groups and the side-chain of Arg38 and the backbone carbonyl of Phe308 were the key interactions for efKAS III inhibition [123].

Both Gram-positive and Gram-negative bacteria produce hyaluronidases, which are an important virulence factor. They enable the bacteria to avoid the immune system and host defense mechanisms. Terpenes (e.g., glycyrrhizin) have been identified as hyaluronic acid lyases (Hyal B from *Streptococcus agalactiae*, Hyal S from *Streptomyces hyalurolyticus*, and Hay C form *Streptococcus equisimilis*). Compounds with many hydroxyl groups inhibited hyaluronate lyase stronger than those with only a few [124].

Flavonoid 5,6-dihydroxy-4′,7,8-trimethoxyflavone, isolated from *Limnophila heterophylla* Benth, was found to effectively kill *Bacillus subtilis* by cell lysis. Moreover, they enhanced the activity of gluconeogenic fructose 1,6-bisphosphatase, but the decreased activity of phosphofructokinase and isocitrate dehydrogenase, the key enzymes of the Embden–Meyerhof–Parnas pathway and the tricarboxylic acid cycle, respectively, was demonstrated [125].

Isoflavones (4-(p-hydroxyphenethyl) pyrogallol and 7,8,4′-trihydroxyisoflavone **(80)** are potent inhibitors of urease, an enzyme produced by *Helicobacter pylori*, which catalyzes the hydrolysis of urea to produce ammonia and carbon dioxide and to protect the bacteria in the acidic environment of the stomach [126]. The structure–activity relationship of these polyphenols revealed that the two *o*-hydroxyl groups were essential for the inhibitory activity of polyphenol. When the C-ring of isoflavone was broken, the inhibitory activity markedly decreased.

It was observed [127] that treatment *Pseudomonas aeruginosa* with cranberry type-A proanthocyanidins **(23,24)** caused downregulation of a wide variety of proteins, including those related to ATP synthesis (likely cytochrome C PA2482), purine, carbohydrate, amino-acid and fatty acid metabolism (HmgA, GuaB, FdhE, FoaB, LdcA, PurU1) and involved in nucleic acid synthesis and repair (e.g., TopA, Rne, RplC, and Mfd). In addition, several citric acid cycle proteins, such as subunits of the acetyl-CoA carboxylase, aconitate hydratase and fumarase, were found to be significantly reduced. However, more than 30 proteins, mainly related to metal cation utilization, were upregulated.

### 4.2. Inhibition of Bacterial DNA Synthesis and Interaction with Nucleic Acids

Flavonoids from *Elaeagnus glabra* were tested for their antibacterial activity against *Proteus vulgaris* and *Staphylococcus aureus*. A free 3′,4′,5′-trihydroxy B-ring and a free 3-OH group were necessary for antibacterial activity. DNA synthesis was predominantly inhabited by the active flavonoids in *P. vulgaris*, whereas RNA synthesis was inhibited in *S. aureus* [128]. The most active inhibitors of DNA synthesis were robinetin **(122)**, myricetin **(114)**, and (−)-epigallocatechin **(14)**. It is probable that the B ring of the flavonoids could intercalate or form a hydrogen bond with the stacking of nucleic acid bases and further lead to the inhibition of nucleic acid synthesis in bacteria. The results of Lou et al. [129] demonstrated that *p*-coumaric acid **(166)** had dual mechanisms of bactericidal activity: disrupting bacterial cell membranes and binding to bacterial genomic DNA leading to inhibition of cellular functions, and ultimately to cell death.

Depolarization of membrane and inhibition of DNA, RNA, and proteins synthesis was observed in *S. aureus* and—in higher concentrations—cell lysis, when treated with flavonoids from *Dorstenia* sp., such as 6,8-diprenyleriodictyol **(106)**, isobavachalcone **(148)**, and 4-hydroxylonchocarpin **(149)** [130].

The synthesis of nucleic acid can be inhibited by polyphenols also through topoisomerase inhibition. Flavonoids are inhibitors of topoisomerases, and it plays an important role in their antimycobacterial activity. Docking studies have proved that quercetin **(111)** effectively binds to the subunit B of DNA gyrase through interaction with residues that are in the Toprim domain of the protein. Due to this activity, it inhibited the growth of *Mycobacterium smegmatis* and *Mycobacterium tuberculosis* [131].

Bandele et al. [132,133] have found that polyphenols may act against topoisomerase II in different a manner; (−)-epigallocatechin gallate **(18)** and (−)-epigallocatechin **(14)** were redox-dependent topoisomerase II poisons, kaempferol **(108)** and quercetin **(111)** were topoisomerase II “poisons”, myricetin **(114)** utilized both mechanisms, while (−)-epicatechin gallate **(18)**, and (−)-epicatechin **(12)** displayed no significant activity. Based on the observation, a set of rules has been formed to predict the mechanism of bioflavonoid action against topoisomerase II: while the C4′-OH in B ring is critical for the compound to act as a traditional poison, the addition of –OH groups at C3′ and C5′ increases the redox activity of the B ring and allows the compound to act as a redox-dependent poison. The second rule is that the aromatic and planar structure of the C ring in the flavonols that includes a C4-keto group allows the formation of a proposed pseudo ring with the C5-OH. Disruption of these elements abrogated enzyme binding and precluded the ability to function as a traditional topoisomerase II poison [132,133].

Although the above studies were conducted with human cells, and flavonoids are assumed to be poisons of human topoisomerase IIα and IIβ, there are some data about flavonoids as inhibitors of bacterial type II topoisomerases: DNA gyrase and topoisomerase IIA (also called topoisomerase IV) [92]. Gyrases are enzymes that modify the DNA topology, and they are present only in prokaryotes, making them an attractive target for antibacterial drugs. DNA gyrase consists of two catalytic subunits; GyrA is responsible for DNA breakage and reunion, while the subunit GyrB contains the ATP-binding site. Coumarins and cyclothialidines are natural products that inhibit the ATPase activity of DNA gyrase by blocking the binding of ATP to subunit GyrB [134]. Plaper et al. [135] demonstrated that quercetin **(111)** inhibits the supercoiling activity of the bacterial gyrase and induces DNA cleavage, and the mechanism is probably based on interaction with DNA. They showed that quercetin **(111)** binds to the 24 kDa fragment of gyrase B of *Escherichia coli* with a K(D) value of 15 µM and inhibits ATPase activity of gyrase B. Its binding site overlaps with the ATP binding pocket and could be competitively replaced by either ATP or novobiocin. The proposed mechanism is that quercetin **(111)** inhibits gyrases through either the interaction with DNA or with the ATP binding site of gyrase [135]. Other polyphenols that can inhibit bacterial DNA gyrase by binding to the ATP binding site of the gyrase B subunit are catechins, with epigallocatechin gallate **(18)** being the most active, followed by epicatechin gallate **(17)** and epigallocatechin **(14)** [136]. Furthermore, quercetin **(111)**, apigenin **(38)**, and 3,3′,4′,6,7-pentahydroxyflavone **(167)** demonstrated inhibitory activity against *Escherichia coli* DNA gyrase [137].

The quantitative structure–activity relationship (QSAR) and molecular docking of flavonoids were analyzed in the study of Fang et al. [138]. The QSAR models demonstrated that hydrophobicity, H-bond donor, steric and electronic properties are key factors for the antibacterial activity of flavonoids. Structure requirements including hydroxyl group at C-3, C-5, C-7 and C-3′, C2-C3 unsaturated double bond and the carbonyl group at C-4 are essential, while the presence of hydroxyl group at C-6, methoxyl group at C-8 and C-3′ could decrease the antibacterial activity. Docking results indicated that half of the tested flavonoids inhibited GyrB by interacting with ATP pocket in the same orientation. Polymethoxyl flavones, flavonoid glycosides, and isoflavonoids changed their orientation, resulting in a decrease in inhibitory activity. Hydroxyl group at C-3, C-5, C-7 and C-4′, carbonyl group at C-4 are key active substituents of flavonoids for inhibiting GyrB by interacting with its key residues. Structure changes, including glycosylation, polymethoxylation or isoflavonoids, will change the action mode and result in a decrease in inhibitory activity [138].

Three flavonoids isolated from cottonseed flour which promoted *Escherichia coli* topoisomerase IV-dependent DNA cleavage were identified as rutin **(121)**, quercetin 3-*O*-rhamnogalactoside, and isoquercetin **(126)**. Moreover, rutin **(121)** also inhibited topoisomerase IV-dependent decatenation activity and induced the SOS response of a permeable *E. coli* strain [139].

Arima et al. [140] observed that morin alone, at a concentration of 50 μg/mL inhibited the synthesis of DNA in the cells of *Salmonella enteritidis*, while its concentration equaled to a concentration of 12.5 μg/mL was enough if rutin **(121)** was added to the medium at a concentration of 12.5 μg/mL. Morin **(110)** alone also inhibited RNA and protein synthesis, but the rutin added did not influence the inhibition process.

Tannic acid **(215)** is strongly bound to DNA, which possibly had led to the covalent modification of DNA bases. Furthermore, tannic acid in the presence of Cu(II) caused strand cleavage in supercoiled plasmid DNA [141].

### 4.3. Interaction with the Bacterial Cell Wall or Inhibition of Cell Wall Formation

Various strengths of antimicrobial activity against bacteria may be caused by differences in cell surface structures between Gram-negative and Gram-positive species [20,64]. The major function of the cell wall is to provide shape and cell integrity and to act as an osmotic barrier. Gram-negative bacteria were reported to be resistant toward many antibacterial substances due to the hydrophilic surface of their outer membrane and associated enzymes in the periplasmic space, which is capable of breaking down many molecules introduced from outside [142,143]. Moreover, the negatively charged lipopolysaccharide (LPS) of the outer membrane protects the bacterial cell against catechins [144]. Gram-positive bacteria seem to be more susceptible to the action of phenolic acids than Gram-negative bacteria [64]. One of the explanations is that the Gram-positive bacterium lacks an outer membrane, which would facilitate diffusion of the phenolic acids through the cell wall and intracellular acidification. Vattem et al. [145] postulated the hyperacidification at the plasma membrane interphase, being a consequence of dissociation of phenolic acids, as one of the possible mechanisms of the antimicrobial action of phenolic acids. This hyperacidification would alter cell membrane potential, making it more permeable, and cause irreversible alterations in the sodium-potassium ATPase pump, therefore leading to cell death.

Wu et al. [146] have demonstrated that quercetin **(111)** and apigenin **(38)** influence the synthesis of the bacterial cell walls by the inhibition of D-alanine:D-alanine ligase (an essential enzyme that catalyzes the ligation of d-Ala–d-Ala in the assembly of peptidoglycan precursors). Moreover, these two flavonoids could inhibit the FabZ enzyme from *H. pylori* [122]. Tasdemir et al. [38] found that quercetin **(111)** could inhibit three consecutive enzymes, β-ketoacyl-ACP reductase (FabG), β-hydroxyacyl-ACP dehydrase (FabZ) and enoyl-ACP reductase (FabI), in the FAS II pathway of *Plasmodium falciparum*, whereas apigenin **(38)** could only inhibit FabI.

Flavones form a complex with cell wall components and consequently inhibit further adhesions and microbial growth as well. The inhibition of bacterial enzymes (such as tyrosyl-tRNA synthetase) was observed for C-7-modified flavonoids containing the naringenin **(81)** core [147]. It was also demonstrated that they were also inhibitors of *S. aureus*, *E. coli*, and *Pseudomonas aeruginosa* growth. Baicalein **(45)** was an effective bactericide, and when combined with cefotaxime, the synergistic effects were observed by inhibiting extended-spectrum β-lactamase CTX-M-1 mRNA expression [148]. Inhibition of the bacterial efflux pump and increase in the susceptibility of existing antibiotics (by inducing depolarization of the cell membrane) is another possible mechanism of antibacterial activity. Artonin I **(63)**, from *Morus mesozygia*, was effective against *S. aureus* due to blocking the efflux mechanism and causing depolarization of the cell membrane [149]. Artonin I **(63)** reversed multidrug resistance and increased the susceptibility of existing antibiotics by lowering their minimum inhibitory concentrations.

Many researchers have proved that polyphenols can interact with the cell wall or outer membrane and with their components such as peptidoglycan, lipopolysaccharide. Zhao et al. [150] demonstrated that unlike dextran and lipopolysaccharide, peptidoglycan from *S. aureus* blocked both the antibacterial activity of EGCG **(18)** and the synergism between EGCG and oxacillin, suggesting EGCG may directly bind to the cell wall of *S. aureus* and interfere with its integrity. These results were confirmed by Yoda et al. [113]. As the bactericidal activity of EGCG **(18)** for *S. aureus* was blocked, dose-dependently by purified peptidoglycan, but not by lipopolysaccharide or dextran, it was suggested that EGCG binds directly to the peptidoglycan in the cell wall. These results are consistent with the opinion that the structure of the bacterial cell wall is responsible for the different susceptibilities of Gram-positive and Gram-negative cells to polyphenols.

Gram-negative bacteria (e.g., *Escherichia coli, Salmonella, Shigella*) have in their outer cell membrane a strong “endotoxin”—lipopolysaccharide (LPS). The fraction of procyanidins from cranberries composed of polymers with an average degree of polymerization of 21 can efficiently bind lipopolysaccharide and prevent the interaction of LPS with receptors on the surface of mammalian target cells [151]. On the other side, phenolic extracts of cloudberry and raspberry rich in ellagitannins disintegrated the outer membrane of examined *Salmonella* sp. [152] and released LPS from bacteria cells.

### 4.4. Alteration of Cytoplasmic Membrane Function

The (inner) bacterial cell membrane is responsible for many essential functions: osmoregulation and respiration processes, transport, biosynthesis and the cross-linking of peptidoglycan and synthesis of lipids [153]. Any disturbance in its structure or functionality can result in metabolic dysfunction and cell death; hence the membrane disruption is postulated to be one of the mechanisms of the antibacterial activity of polyphenols. For example, catechins were shown to rupture the bacterial membrane by binding to the lipid bilayer and by inactivating or inhibiting the synthesis of intracellular and extracellular enzymes [154].

Apigenin **(38)** induced fungal membrane dysfunction and increased cell permeability [155], which caused the release of small intracellular constituents such as ions and sugars, but not proteins. Epicatechin-3-gallate **(17)** and caffeic acid **(167)** targeted both the cell wall and cytoplasmic membrane of *P. aeruginosa* [156]. The cellular membrane destruction and ensuing membrane permeability perturbation of *P. aeruginosa* had led to the ascending access of hydrophobic antibiotics, a release of potassium ions, and leakage of nucleotides. Phenolic acids, due to their partially lipophilic nature, pass through the cell membrane by passive diffusion and cause an increase in membrane permeability. They possibly reduce the intracellular pH and induce protein denaturation [157]. Methanol extract of *Coriolus versicolor* rich in polyphenols disabled *S. aureus* cell division (i.e., the formation of septa) and led to the accumulation of peptidoglycan and teichoic acid precursors in the cytoplasm [157]. In this case, the extract acted directly on the cytoplasmatic membrane, whereas in Gram-negative *Salmonella* Enteritidis, the cell envelope was damaged. On the other side, at high concentrations, catechins were found to generate an oxidative burst by the generation of reactive oxygen species (ROS) that cause alteration in the membrane permeability and membrane damage [158]. Purified flavonoids from *Graptophyllum glandulosum* possessed antimicrobial activities against multidrug-resistant *Vibrio cholerae* and caused cell lysis and disruption of the cytoplasmic membrane upon membrane permeability [159].

Flavonoids (acacetin **(40)** and apigenin **(38)**) and flavonols (morin **(110)** and rhamnetin **(115)**) caused destabilization of the membrane structure by disordering and the disorientation of the membrane lipids and induced leakage from the vesicle [160]. The inverse correlation between the number of hydroxyl groups in the flavonoids and their capacity to leakage induction was noted. Studies of Chabot et al. [161] suggested that flavonoids lacking hydroxyl groups on their B rings (genistein **(65)**, hesperetin **(83)**, chrysin **(47)**, galangin **(109)**) were more potent inhibitors of microbial growth than those with the –OH groups. On the other hand, Adamczak et al. [67] have demonstrated that the presence of hydroxyl groups in the phenyl rings A and B usually did not influence the level of the antibacterial activity of flavones. A significant increase in the activity of the hydroxy derivatives of flavone was observed only for *S. aureus*. What is interesting, in contrary to other studies, the compounds tested in the study were generally more active against Gram-negative bacteria: *Escherichia coli* and *Pseudomonas aeruginosa* than Gram-positive ones: *Enterococcus faecalis* and *Staphylococcus aureus*.

Tsuchiya [162] reported that the catechin impact on membrane fluidity also depended on the stereospecificity. (−)-Epicatechin **(12)**, (+)-epicatechin **(9)**, (−)-catechin **(10)** and (+)-catechin **(7)** reduced membrane fluidity in increasing order of intensity; it means that epicatechins in a *cis* form were more effective for reducing membrane fluidity than catechins in a *trans* form. Stereospecificity in the membrane effects of catechin stereoisomers may be induced by the different hydrophobicity of geometrical isomers and the chirality of membrane lipid components. Lipophilic flavonoids may also disrupt microbial membranes [107]. It was suggested that the mode of action of terpenes and their related alcohols involves disruption of microbial membranes by their lipophilic components [163]. According to Tsuchiya [164], bioactive components with amphiphilic or hydrophobic structures interact with biological membranes resulting in the modification of membrane fluidity, microviscosity, order, elasticity, and permeability. The author postulated that interactions of flavonoids with lipid bilayers involve two mechanisms; the first is associated with the partition of the more nonpolar compounds in the hydrophobic interior of the membrane, while the second one includes the formation of hydrogen bonds between the polar head groups of lipids and the more hydrophilic flavonoids at the membrane interface. The membrane interactions and localization of flavonoids play a vital role in altering membrane-mediated cell signaling cascades [165].

The studies of Arora et al. [166] demonstrated that flavonoids and isoflavonoids preferentially enter into the hydrophobic core of membranes. In plant tissues, flavonoids occur mainly in the form of glycoside, and the presence of glycosidic residues in the flavonoid skeleton influences the hydrophobicity of flavonoids. Flavonoids with a greater hydrophobicity have been reported to influence the transmembrane potential to a greater extent than the less hydrophobic flavonols, probably because they can enter deeper into the lipid bilayer, thereby disrupting the compact packing of lipids [157]. Moreover, the spatial configuration is also important; a substantially higher affinity for artificial membranes was reported for flavonols (planar) than flavanones (tilted) [167]. Using the fluorescence anisotropy technique, it was reported that naringenin and naringin enhanced membrane fluidity, while membrane interaction with quercetin **(111)**, daidzein **(64)**, luteolin **(49)**, galangin **(109)**, kaempferol **(108)** and genistein **(65)** resulted in rigidified membranes [165]; however, the impact depended on the lipid composition of membranes.

Wu et al. [168] have shown the positive correlation between antibacterial capacity and membrane rigidification effect of the polyphenolic compounds. Authors have observed that flavonoids decreased the membrane fluidity with the potency being kaempferol **(108)** > chrysin **(47)** > baicalein **(45)** > quercetin **(111)** > luteolin **(49)**, whereas isoflavonoids increased the membrane fluidity with the potency being puerarin **(70)** > ononin **(75)** > daidzein **(64)** > genistin **(68)** [168]. Kaempferol **(108)**, located deeply in the hydrophobic core of the lipid bilayer, decreased the membrane fluidity most and exhibited the highest antibacterial capacity against *E. coli*. The number and the position of hydroxyl groups influenced the membrane interaction with polyphenols; the OH group at C-3 in the C ring was important for decreasing membrane fluidity. He et al. [169] suggested that for flavonoids to be effective antimicrobial agents, interaction with the polar head–group of the model membrane followed by penetration into the hydrophobic regions must occur. The antimicrobial efficacies of the flavonoids were consistent with liposome interaction activities and decreased in the order: kaempferol **(108)** > hesperetin **(83)** > (+)-catechin **(7)** > biochanin A **(76)**.

Sophoraflavanone B (8-prenylnaringenin) **(101)** caused cell wall weakening, membrane damage and intracellular constituents leaking from the cell of methicillin-resistant *S. aureus* [170]. In the study, the direct binding of sophoraflavanone B **(101)** to peptidoglycan was demonstrated. It also has been proposed that sophoraflavanone G **(98)** and (−)-epigallocatechin gallate **(18)** inhibited cytoplasmic membrane function [171].

### 4.5. Inhibition of Energy Metabolism

Many aspects of cellular metabolism revolve around ATP production and consumption, and ATP is regarded as the universal energy exchange factor that connects anabolism and catabolism but also enables processes such as motile contraction, phosphorylation, and active transport (uptake) of nutrients. Membrane-bound F1F0 ATP synthase from bacteria is an enzyme responsible for ATP production through oxidative phosphorylation or photophosphorylation. It has been demonstrated that morin **(110)**, baicalein **(45)**, silibinin **(138)**, and epicatechin caused complete inhibition of ATPase activity, while hesperidin **(86)**, chrysin **(47)**, kaempferol **(108)**, diosmin **(51)**, apigenin **(38)**, genistein **(65)**, or rutin **(121)** exert partial inhibition of about 40–60% [172].

Dadi et al. [173] demonstrated that resveratrol **(202)**, piceatannol **(203)**, quercetin **(111)**, quercitrin **(123)**, or quercetin-3-β-D glucoside **(126)** inhibited *E. coli* ATP synthase, but to different degrees. The most potent inhibitor was piceatannol **(203)** (~0 residual activity); inhibition by other compounds was partial and ranged from ~20% residual activity for quercetin **(111)** to ∼60% residual activity for quercitrin **(123)** or resveratrol **(202)**. Inhibition was identical in both F_1_F_0_ membrane preparations as well as in isolated, purified F_1_, but in all cases, inhibition was reversible. Interestingly, resveratrol **(202)** and piceatannol **(203)** inhibited both ATPase and ATP synthesis, whereas quercetin **(111)**, quercitrin **(123)** or quercetin-3-β-D glucoside **(126)** inhibited only ATPase activity and not ATP synthesis. The membrane-bound ATPase activity of *E. coli* was also inhibited by eugenol **(185)** or carvacrol **(183)** [174]. Similar results were obtained for thymoquinone that completely inhibited both purified F_1_ and membrane-bound F_1_F_0_
*E. coli* ATP synthase, and the process of inhibition was fully reversible [175].

It has been proposed that licochalcones A **(150)** and C **(151)** can inhibit energy metabolism [171]. Haraguchi et al. [176] have proved the antibacterial activity of licochalcones A **(150)** and C **(151)** against *S. aureus* and *Micrococcus luteus*, which resulted from inhibited oxidation of NADH in bacterial membranes. As licochalcones inhibited NADH-cytochrome c reductase, they exerted their antibacterial activity by inhibiting the bacterial respiratory electron transport chain.

EGCG **(18)** directly interacts with proteins and phospholipids in the plasma membrane and regulates signal transduction pathways, transcription factors, DNA methylation, as well as mitochondrial function and autophagy [177].

### 4.6. The Inhibition of Biofilm Formation and Interfering with Bacterial Quorum Sensing

Biofilm is an assemblage of microbial cells that are irreversibly linked to a surface with bacteria embedded in an extracellular matrix of self-produced biopolymers. The ability to form biofilm is an important property of various bacteria such as pathogenic species and is associated with quorum sensing (QS) or cell-to-cell communication. Therefore, bacterial cell-to-cell communication has received attention to manifest the role of quorum signals in the attachment and growth of pathogenic bacteria in foods. QS participates in the biofilm formation as well as controls the expression of various virulence factors, inter alia, the production of proteases that degrade connective tissue, the production of siderophores that facilitate iron uptake, the releasing of toxins that disrupt cellular processes, the formation of phenazines that favor the reactive oxygen species generation, and production of exopolysaccharides that are necessary for the phagocytosis-resistant capsules structure [178].

An anti-quorum sensing (anti-QS) agent curcumin **(216)** from *Curcuma longa* (turmeric) was shown to inhibit the biofilm formation of pathogens, such as *Escherichia coli*, *Pseudomonas aeruginosa* PAO1, *Proteus mirabilis* and *Serratia marcescens*, possibly by interfering with their QS systems [179], because the biofilm maturation was disturbed by a biomass reduction and by the interruption of swimming motility. Chlorogenic acid **(172)** was proved to significantly inhibit the formation of biofilm by *P. aeruginosa*, its ability to swarm, and virulence factors including protease and elastase activities and rhamnolipid and pyocyanin production. Moreover, the QS related genes were downregulated in *P. aeruginosa*, and the inhibitory rates were as follows: lasI 85.09%, lasR 48.63%, rhlI 27.98%, rhlR 34.7%, pqsA 73.08%, and pqsR 45.85%, respectively [180].

Quercetin **(111)** efficiently reduced the biofilm formation and other QS regulated phenotypes like violacein inhibition, exopolysaccharide production and alginate production in foodborne pathogens *K. pneumoniae*, *P. aeruginosa*, and *Y. enterocolitica* [181]. Furthermore, quercetin **(111)** significantly inhibited the swimming and swarming behavior of *P. aeruginosa* and *Y. enterocolitica*.

*L. paracasei* exposed to resveratrol **(202)** displayed changes in the physicochemical properties of their surface, especially with a global increase in negative charges, a more basic nature and an increase in their hydrophobicity. These changes may largely contribute to the enhanced adhesion, induced formation of bacterial aggregates and biofilm formation abilities of resveratrol-treated *L. paracasei* [182]. However, the majority of studied polyphenols have shown the opposite impact on the ability of biofilm formation. Apple flavonoid phloretin **(152)** was reported to control *E. coli* O157:H7 biofilm formation by a mechanism that implies repressing the curli genes (csgA and csgB), which are involved in fimbriae production [183]. Epigallocatechin-3-gallate **(18)** eliminates the biofilm matrix by directly interfering with the assembly of curli subunits into amyloid fibers and by triggering the σ^E^ cell envelope stress response and thereby reducing the expression of a crucial activator of curli and cellulose biosynthesis (CsgD) [184]. Recently, it has been shown that EGCG **(18)** act against biofilms by strongly interfering with the assembly of amyloid fibers and the production of phosphoethanolamine-modified cellulose fibrils [185].

EGCG **(18)** also inhibited the formation of *Streptococcus mutans* biofilms [186,187]. The growth of *Streptococcus mutans* decreased, and the biofilm formation was inhibited by pinocembrin **(95)**, apigenin **(38)**, quercetin **(111)**, while caffeic acid phenethyl ester decreased, probably due to changes in bacterial architecture [188].

According to Xu et al. [189], the ECGC **(18)** activity against *S. mutans* is due to disrupting at the transcriptional level the adherence of bacteria to surfaces and hence inhibiting the biofilm formation. Authors hypothesized that EGCG at sublethal concentrations directly suppressed the expression of gift genes encoding glucosyltransferases, enzymes that synthesize polysaccharides necessary for biofilm formation [189]. Moreover, EGCG **(18)** was found to inhibit the enzymatic activity of the F_1_F_0_-ATPase and lactate dehydrogenase [190].

Morin **(110)** at its sub-MICs demonstrated a significant dose-dependent inhibitory efficacy against *Listeria monocytogenes* biofilm formation [191]. Moreover, morin-treated *Listeria* showed a significant reduction in hemolysin secretion and a concentration-dependent decrease in the flagella directed swimming and swarming velocity. The biofilm formation and biofilm-related genes in *L. monocytogenes* also were inhibited by thymol **(184)**, carvacrol **(183)** and eugenol **(185)** [192].

Bap (biofilm-associated protein) is expressed by *Staphylococcus* sp. in order to adopt functional amyloid-like structures as scaffolds of the biofilm matrix. Quercetin **(111)**, myricetin **(114)** and scutellarein **(44)** specifically inhibited Bap-mediated biofilm formation of *S. aureus* and other staphylococcal species [97] by preventing the assembly of Bap-related amyloid-like structures.

The ability of bacteria to adhere was also inhibited by phenolic acids. Adhesion was less favorable when the bacteria were exposed to gallic acid **(159)** (*P. aeruginosa*, *S. aureus* and *L. monocytogenes*) and ferulic acid **(168)** (*P. aeruginosa* and *S. aureus*). Both phenolics were able to inhibit bacterial motility and prevented biofilm formation, as well as reducing the mass of biofilms formed by the Gram-negative bacteria [193]. Further studies proved that gallic **(159)** and ferulic **(168)** acids led to irreversible changes in membrane properties, such as its charge, intra and extracellular permeability, and physicochemical properties. Both acids caused changes in hydrophobicity and negative surface charge and induced the local rupture or pore formation in the cell membranes leading in consequence to essential intracellular constituent leakage [194].

### 4.7. Substrate Deprivation

When polyphenol forms complex with protein, the biological function might change. Depending on the function of a complexed protein, the influence on bacteria cells will differ. As mentioned above, a decreased or inhibited activity of enzymes can result in a lack of energy for bacteria and, in consequence, might lead to cell death. Lack of energy also means a disturbed transport of nutrients across the cell wall and cytoplasmic membranes, diminished bacteria proliferation and limited mobility, as well as inhibited ability to biofilm formation or even blocked sporulation [195].

The deprivation of the substrates required for microbial growth, especially essential mineral micronutrients such as iron and zinc (via proanthocyanidin chelation with the metals), together with the destabilization of the cytoplasmic membrane, the permeabilization of the cell membrane, the inhibition of extracellular microbial enzymes, direct actions on microbial metabolism, were supposed to be the mechanism of the antibacterial activity of the A-type proanthocyanidin **(23,24)** [196].

Scalbert [197] suggested that tannin toxicity for bacteria is due to the direct impact on bacterial metabolism by inhibiting the oxidative phosphorylation as well as by deprivation of the substrates required for microbial growth, especially an iron deprivation. Generally, tannins are reported to be strong inhibitors of many various hydrolytic enzymes such as α-amylase, pectinase, cellulase, xylanase, lactate dehydrogenase, malate dehydrogenase, peroxidase, β-glucosidase, so they can inhibit the activity, growth or proliferation of microorganisms [198]

The inhibitory effect of tannic acid **(215)** on the growth of intestinal bacteria may be due to its strong iron-binding capacity. The growth of *E. coli* was restored by the addition of iron to the medium after the precipitate caused by tannic acid **(215)** was removed [199]. In the study, neither *Bifidobacterium infantis* nor *Lactobacillus acidophilus* required iron for growth, which probably contributes to their resistance to tannic acid. It is known that only a few bacteria, including lactobacilli, do not require iron. It is an essential trace element for most gut bacteria, and many have active Fe transport systems and other mechanisms to scavenge Fe [200]. For example, *Bacteroides* spp. are highly dependent on heme and iron, whereas many members of the Enterobacteriaceae have developed mechanisms, including siderophores, to acquire Fe in competition with other bacteria and the host.

It is well known that catechol and gallol structure (Figure 3) and hence many polyphenolic compounds are effective metal chelators. After deprotonation, which is required for metal binding, catecholate and gallate groups may be complexed with metal ions that prefer octahedral geometry, such as Fe^2+^ and Fe^3+^ (Figure 3) [201].

It was established for flavones and for the flavanone naringenin **(81)** that the binding metal sites are preferentially at the 5-hydroxyl and 4-oxo groups [202]. On the other hand, the study of Mladenka et al. [203] demonstrated that the most effective iron-binding site of flavonoids is 6,7-dihydroxy structure, present, for example, in baicalein **(45)**. The simultaneous presence of 3-hydroxy-4-keto conformation, 2,3-double bond and the catecholic B ring were associated with significant iron chelation; however, the catecholic B ring did not play an essential role in more acidic conditions. Quercetin **(111)** and myricetin **(114)** that contain all mentioned structural requirements, had activity similar to baicalein **(45)** at the neutral conditions but were clearly less active in lower pH. On the other hand, baicalein **(45)**, additionally possessing the 6,7-dihydroxyl groups, was very efficient even in the acidic condition. The 5-hydroxy-4-keto configuration has only moderate activity at all pH conditions. It was also proved that isolated keto, hydroxyl, methoxyl groups or ortho methoxy–hydroxy groups were not associated with iron chelation at all.

Polyphenols also have strong binding interactions with Cu^2+^, and stability constants for Cu^2+^ catecholate complexes are even larger than for Fe^2+^ [201].

As bacteria contain various metalloenzymes, flavonoids by binding metal ions can inhibit their activity and lead to various metabolic disorders (enzyme inhibition, impairment of ion channel functions). Furthermore, the metabolic functions of the human gut microbiota that involve metalloenzymes may also be altered [204]. An enzyme methionine aminopeptidase (MetAP) carries out the removal of the initiator methionine residue from newly synthesized proteins, and this removal is critical for the activation, distribution and stability of many proteins. It was proved that the adjacent hydroxyl groups on the phenyl ring (catechol moiety) were essential for effective inhibition of the Fe (II)-a form of *E. coli* MetAP and growth inhibition of bacterial cells [205].

Polyphenols can also cause iron deficiency in the digestive tract, which will affect sensitive bacterial populations and change the composition of the intestinal microbiota. Oral bacterium Fusobacterium nucleatum is associated with colon cancer, causes erythrocytes lysis, and therefore releases hemoglobin, which provides an iron source to bacteria and other periodontopathogens, promoting their proliferation in periodontal pockets [206]. The tea polyphenols were proved to inhibit dose-dependently the hemolytic activity of *F. nucleatum*.

The virulence factors such as gelatinase, collagen-binding antigen, cytolysins, and proteases enhance colonization, survival and persistence of *E. faecalis* in the root canal. Treatment of *E. faecalis* with a sublethal concentration of EGCG **(18)** (2.5 mg/mL) significantly inhibited the expression of responsible genes (collagen adhesin (ace), cytolysins activator (cylA), gelatinase (gelE) and serine protease (sprE)) by >75% compared to the untreated control [207]. The elastase, protease and pyocyanin production in *P. aeruginosa* were inhibited by curcumin **(216)** in a dose-dependent manner [179]. EGCG **(18)** caused the inhibition of glucose uptake by *E. coli*, which can suggest that EGCG inhibits the major function of porin proteins, namely the passive transport of small hydrophilic molecules such as glucose, leading to growth inhibition of *E. coli* [114].

### 4.8. The Relationship between Polyphenols Structure and Antibacterial Activity

The mechanism of inhibition by polyphenols may differ depending both on the structure of the polyphenolic compound and bacteria species. The amphipathic character of flavonoids plays a very important role as hydrophilic and hydrophobic moieties must be present together and well-spaced in these compounds [208].

Flavans with prenyl group at the A ring were potent antibacterial compounds against *Staphylococcus aureus*, and the number and position of prenyl groups on this ring influenced the activity [91].

The number of hydroxyl groups in the B ring in flavonols and flavones is associated with the antimicrobial activity against lactic acid bacteria (LAB). Myricetin **(114)** is a flavonol possessing three hydroxyl groups in the B ring as pyrogallol structure, whereas quercetin **(111)** and kaempferol **(108)** have one and two hydroxyl groups less in the B ring than in myricetin, respectively. Myricetin, as a pure compound, significantly inhibited the growth of all tested LAB that originated from the human gastrointestinal tract, as well as the Gram-positive *Enterococcus faecalis* and *Bifidobacterium lactis*, while quercetin **(111)** and kaempferol **(108)**, with a more lipophilic nature, had no inhibitory impact on the above bacteria [47]. Flavone luteolin **(49)** has a structure similar to quercetin **(111)**, but it lacks the OH group at position 3 in ring C. Luteolin **(49)** was bacteriostatic against some of the tested LAB as well as against *E. faecalis* and *B. lactis*, while other flavone apigenin **(38)**, which has one hydroxyl group less in the B ring had no such effects.

Baicalein (flavon) **(45)** and myricetin (flavonol) **(114)** show the most significant antibacterial effects among the tested flavonoids. Both have a pyrogallol structure, but baicalein in ring A (5, 6, 7–OH) and myricetin in ring B. Results proved that the pyrogallol structure was an important element for the potent antibacterial activity for flavonoids [60]. Echeverria et al. [208] made the comparison between a flavone (planar) and flavanone (not planar) with similar lipophilicity and oxygenated substitution patterns in the A and B rings (e.g., pinocembrin **(95)** and 3-*O*-methylgalangin **(113)**) and showed that flavones have higher antibacterial activity. On the other hand, possessing at least one hydroxy group in the ring A (especially at position C-7) seems to be crucial for antibacterial activity of flavones, and an additional OH group in another position such as C-5 and C-6 can further increase the activity [91].

All the flavonols and flavanones with antibacterial activities had two hydroxyl substituents on C-5 and C-7 of ring A in common, such as quercetin **(111)**, rutin **(121)**, naringenin **(81)**, and hesperetin **(83)** [60]. Moreover, the authors suggest that flavanones were more active than the corresponding flavones. For example, naringenin **(81)** showed antibacterial effects on all the tested bacteria, whereas apigenin **(38)** showed almost no effect. Such results indicate that the saturation of the C2-C3 double bond increased the antibacterial activity.

On the other side, Wu et al. [168] demonstrated that flavonoids were more effective *E. coli* inhibitors than isoflavonoids with relative activity being as follows: kaempferol **(108)** > quercetin **(111)** > chrysin **(47)** > luteolin **(49)** > baicalein **(45)** > tangeretin **(57)** and daidzein **(64)** > genistin **(68)** > ononin **(75)** > puerarin **(70)**. The only structural difference between quercetin **(111)** and luteolin **(49)** is that quercetin has a hydroxyl group at position 3 in the C ring, while luteolin has none. It means that the 3-OH group is important to the activity of flavonoids against Gram-negative bacteria *E. coli*. Further analysis of structure–activity relationships revealed that the methylation of OH groups could decrease the antimicrobial activity of flavonols. It also has been shown a significant positive correlation between the antibacterial capacity of flavonoids and the membrane rigidification effect. A quantitative structure–activity relationship (QSAR) study revealed that the activity of the flavonoid compounds could be related to molecular hydrophobicity and charges on the C atom at position 3 [168].

The hydrophobic substituents such as prenyl groups, alkylamino chains, alkyl chains, and nitrogen or oxygen-containing heterocyclic moieties usually enhance the antibacterial activity for all the flavonoids [98]. It was concluded that hydroxyl groups on special sites are favorable for antibacterial activity, such as 5,7-dihydroxyl substitution for flavone and flavanone and 2′ or 4′ hydroxylation for chalcones. The hydroxyl group at position three on the C ring of flavone also increased the activity. However, the methylation of the hydroxyl groups generally decreased the activity. The lipophilicity of ring A is therefore of great importance for the activity of chalcones. In addition, hydroxy groups at 4′, 4, and 6 of A and B rings increase the activity of chalcones [91].

The substitution of the flavonoid ring system with prenyl groups increases the lipophilicity of the molecule and results in a strong affinity to biological membranes. Prenylated flavonoids, i.e., featuring C_5_ isoprenoid substituents, have a relatively narrow distribution in the plant kingdom and are constitutively expressed in plants, as compared with prenylated isoflavonoids, which are produced in response to an attack or damage [209]. Xanthohumol **(145)** is the main component (80–90% of the total flavonoids) and is the most abundant prenylated chalcone in hops. It exerted high antimicrobial activity against *Bacteroides fragilis, Clostridium perfringens* and *Clostridium difficile* [210]. β-bitter acids (lupulones) were less effective, and the least effective against anaerobic pathogens were α-bitter acids (humulones). Xanthohumol **(145)**, naringenin **(81)**, chalconaringenin **(140)** and 4-hydroxy-4′-methoxychalcone inhibited the growth of *S. aureus* [211]. The presence of at least one hydroxyl group and especially at the C-4 position was crucial for the antibacterial activity against *S. aureus*. The lack of hydroxyl group or its replacement by a halogen atom (–Cl, –Br), nitro group (–NO_2_), ethoxy group (–O–CH_2_CH_3_), or aliphatic groups (–CH_2_CH_3_), (–CH_3_) led to inactivation of the compounds. Prenylated flavonoids, such as artocarpin **(62)** and isobavachalcone **(148)**, exhibited strong antibacterial activity towards *B. cereus*, *E. coli*, and *Pseudomonas putida* or only Gram-positive species, respectively [212]. It has been demonstrated that any isoflavonoid modification that results in the absence or cyclization of the prenyl group decreases the antibacterial activity of the compound.

Campos et al. [213] had demonstrated that hydroxycinnamic acids (p-coumaric **(166)**, caffeic **(167)** and ferulic **(168)** acids) induced greater potassium and phosphate leakage than hydroxybenzoic acids (protocatechuic **(157)**, gallic **(159)**, and vanillic **(161)** acids) across the membranes of *Oenococcus oeni* and *Lactobacillus hilgardii*.

Flavonoids can occur in two forms: free as “aglycons” or in the form of “glycosides”, where an aglycon is combined with sugar moiety (“glycone”). Flavonoid glycosides occur in a diet generally in ring A or C as O-glycosides, and a corresponding substitution in ring A has a far greater impact on activity [180]. Aglycones of most flavonoids are more hydrophobic than their glycosides [91]. Both the number of glycosylation as well as the position and structure of saccharides are of great significance for the antioxidant, antibacterial, anticancer, anti-inflammatory and antidiabetic activity of a compound [180]. It has been postulated that glycosylation of flavonoids enhances antimicrobial activity, but their antioxidant, anti-inflammatory, anticancer and cardioprotective properties decreased [180]. However, it seems that the impact of glycosylation on antibacterial activity depends on the flavonoid class as well as the position at which sugar moiety is added. The results of Duda-Chodak [63] demonstrated that flavonoid aglycones, but not their glycosides, may inhibit the growth of some intestinal bacteria. In this study, rutin (quercetin 3-*O*-rutinoside) **(121)** had no inhibitory influence on the intestinal bacteria analyzed, and even slight stimulation of the growth of *Lactobacillus* spp. was observed. In contrast, its aglycone quercetin **(111)** exerted a dose-depended inhibitory effect on intestinal bacteria (except on *Bifidobacterium catenulatum*), and this was especially strong on *Ruminococcus gauvreauii*, *Bacteroides galacturonicus* and *Lactobacillus* spp. growth. The same was true for flavanones. Naringin **(85)** and hesperidin (flavanone 7-*O*-glycosides) **(86)** had no impact, but their aglycones (naringenin **(81)** and hesperetin **(83)**, respectively) inhibited the growth of almost all bacteria analyzed. A similar result, showing that 7-*O*-glycosylation of flavanones (naringenin and hesperetin) and flavones (baicalein **(45)**) decreased the antimicrobial activity against *E. coli*, *S. aureus*, *S. typhimurium*, *Enterobacter sakazakii* and *Vibrio parahemolyticus* were demonstrated by Xie et al. [60]. The opposite results were obtained by Adamczak et al. [67]; flavonol aglycones kaempferol **(108)** and quercetin **(111)** displayed a moderate activity only against *E. coli*, while quercetin 3-*O*-rutinoside **(121)** demonstrated inhibitory influence on all strains tested.

Docking results have revealed that the substitution of galloyl or glycosides at position 3 of heterocyclic pyrane ring in flavonoids enhanced the binding affinity to three targets, i.e., fumarate reductase flavoprotein subunit (FrdA), dihydroorotate dehydrogenase (PyrD) and NADH-dependent enoyl-ACP reductase (FabI). Such a phenomenon was observed for flavonoids and their glycosides; quercetin 3-rhamnoside **(123)** and myricetin 3-galactoside **(127)** were more potent inhibitors to PyrD, FabI, and DYR than quercetin **(111)** and myricetin **(114)**, respectively [121]. One of the most potent bacterial inhibitors among flavan-3-ol is EGCG **(18)**, possessing both pyrogallol and galloyl structures in a moiety. EGCG is a stronger inhibitor of pathogens than other flavan-3-ols having fewer or no galloyl groups and pyrogallols. Antibacterial activity of tea flavan-3-ols was in decreasing order EGCG **(18)** > ECG **(17)** > EC **(12)** ≥ theaflavins ≥ gallic acid **(159)** > EGC **(14)** against *S. aureus* and *P. aeruginosa* [214]. The importance of these free galloyl groups for antibacterial activity was also proved in the study of Puljula et al. [215]. Salicarinin A **(213)** and rugosin D **(211)** possess many free galloyl groups, inhibited the growth of *S. aureus* completely at a 0.5 mM concentration. Other ellagitannins, with lower numbers of galloyl or pyrogallol substituents, were less effective.

### 4.9. The Impact of Food Matrix on Polyphenol Activity

There are large discrepancies between the results, i.e., in some studies, it is shown that a given polyphenol class inhibits bacteria, and in others, it does not affect or even stimulates their growth. This may be due to the structure of the used polyphenol, including the molecule size, number and position of hydroxyl groups, their substitutions, the presence/absence and position of glycosylation, hydrophobicity and hydrophilicity of the moiety, and others. The observed discrepancies could also be attributed to the changes in the structure of polyphenols when dissolved in various solvents (water, ethanol, methanol, organic solvents and their mixtures) or after their addition to the medium with bacteria. It is because polyphenols do not dissolve in every solvent, and they can precipitate (affecting the actual concentration of the tested compound) after changing the solvent [216,217,218]. Polyphenols also have different rates of diffusion depending on the medium and environmental conditions.

Moreover, the type of microorganism (Gram-positive, Gram-negative, anaerobic or aerobic or microaerophile, etc.) also has a significant impact when the activity of polyphenols is assessed. It should be borne in mind that used assays, analytical methods, as well as conditions and incubation time, strains of microorganisms, inoculum size, and even concentrations of tested polyphenol may differ between scientific laboratories. There are also big differences between results when the impact of polyphenol on intestinal bacteria is assessed using pure polyphenols solution, plant extract containing polyphenols mixture of whole food in which polyphenols are bound to a food matrix. When in vitro studies are performed, usually pure cultures of bacteria are tested, and interactions with other members of gut microbiota, the impact of human digestive enzymes, the host health, or interactions with other components of a meal are not taken into account. However, all mentioned factors are important for the final results.

It is obvious that the results obtained from in vivo and in vitro studies should not be compared directly. When in vivo studies are conducted, the scientists introduce an ingredient into the diet and analyze changes in the abundance or composition of the gut microbiota, usually focusing on the effect on the entire bacterial population rather than on individual species. In such experiments, many factors contribute to the final results: the chemical composition of the food matrix, the bioaccessibility of polyphenols, their bioavailability, the interactions between particular bacterial strains present in the gut, the health of consumers and many more. Depending on the polyphenols present in the plant, different effects can be achieved because each polyphenol reacts differently with the components of plant tissues. Moreover, each plant differs in its composition. Tarko and Duda-Chodak [219] proved the differences between the bioaccessibility of polyphenolic compounds originating directly from fruits (black chokeberry, elderberry, hawthorn, Cornelian cherry, apple and Japanese quince) and that of those present in the fruit extracts during their digestion conducted in a simulated human gut. They proved significant differences in polyphenols bioavailability that resulted from their interactions with food matrixes. It was caused by polyphenols bounding to the matrix, which is known to modify the polyphenols extractability and susceptibility to digestive enzymes and bacterial metabolism [220]. The interaction with the food matrix also modulates the impact of polyphenols on bacteria inhabiting the colon.

During in vivo studies, it should also be considered that some polyphenols present in a diet are absorbed before they reach the colon, and hence, do not influence the microbiota. For example, quercetin glycosides can undergo partial hydrolysis by pepsin during their passage through the stomach [221], and the released aglycone quercetin **(111)** may be then absorbed in the stomach and secreted in the bile. Glycosides of other flavonoids can be hydrolyzed to aglycones in the small intestine due to the activity of human digestive enzymes, such as lactase phlorizin hydrolase and cytosolic β-glucosidase. It refers to the glycosides that contain glucose, xylose or galactose; as mentioned, humans enzymes e have an affinity for those sugars. It means that only polyphenols resistant to the action of human enzymes are not absorbed in the small intestine and pass to the colon, where they may exert their inhibitory or stimulatory activity towards microbiota, or they may be cleaved by bacterial enzymes to produce derivatives and metabolites of various activity.

Another important issue is the diversity of the chemical composition of plant tissues. For example, chokeberries and apples contain much higher amounts of pectin than the elderberry fruit, which resulted in small amounts of polyphenols in the sediment obtained after elderberry digestion [219]. Further, fruits of the Cornelian cherry are rich in pectin and also in low-molecular-weight phenolic acids that can firmly bind with pectin and so pass to the colon intact [219]. However, the differences between the food matrix could also be related to the cell wall composition of the fruit, resulting in an observed different bioaccessibility of polyphenols present in apples, chokeberries and Japanese fruit [219]. The flesh of Japanese quince fruits contains much pectin, whereas, in the cell walls, cellulose dominates [222]. On the other hand, apples are rich both in pectin and cellulose, but they also contain lignin [223]. The presence of lignins was believed to reduce the proanthocyanidin adsorption in skin cell walls when compared to that of the flesh cell walls [224], causing that unbound proanthocyanidins were more sensitive to enzymatic digestion and acidic pH in the stomach.

Proanthocyanidins are of neutral charge, so they are easily absorbed by the cell wall polysaccharides, while anthocyanins—which are positively charged molecules—could rather selectively bind to a negatively charged pectin [225]. The ratio of bound to free proanthocyanidins depends mainly on their concentration and degree of polymerization. The susceptibility of anthocyanins, anthocyanidins, and proanthocyanidins to digestion can also depend both on the structure of the cell wall polysaccharide network in fruits and the structure of pectin. Voragen et al. [226] have demonstrated that 47% of the structural elements of pectin in apples are neutral side chains, while in bilberry or black currant, more than 60% are homogalacturonan]. Yet another structure was reported for Japanese quince pectin, which consisted of four different populations, mainly arabinans and highly methylated homogalacturonans [227]. The simultaneous presence of pectin, cellulose and hemicellulose in food favors the bounding of procyanidins and anthocyanins and protects them against digestive enzyme activity. In consequence, they are not released from the food matrix at this digestion stage. Moreover, during proanthocyanidins degradation, free (+)-catechin **(7)** could be released, which can bind effectively to cellulose [228].

Tarko and Duda-Chodak [219] also revealed that procyanidin B1 in hawthorn was almost insensitive to digestive enzymes, and probably the saponins, which presence in the hawthorn fruit is characteristic, had such a protective impact. Saponins are poorly absorbed in the intestine mainly due to their unfavorable physicochemical traits, such as large molecular mass (>500 Da), high hydrogen binding capacity (>12), and high molecular flexibility (>10).

Concluding, the presence/absence of the food matrix, as well as its chemical composition, can affect the bioaccessibility, bioavailability and biological activity of polyphenols and their bidirectional interactions with the intestinal microbiota.

## 5. Polyphenols Biotransformation by Intestinal Bacteria

It is believed that only undigested and unabsorbed polyphenols can reach the large intestine and exert their impact on bacteria inhabiting there. As described above, many of the polyphenols can inhibit the growth of microbiota residing in the colon. However, some of the phenolic compounds act as prebiotics and stimulate the growth of particular species. Hence, polyphenols modulate the composition of human gut microbiota. On the other hand, only unabsorbed polyphenols can undergo biotransformation during the activity of bacterial enzymes. Products of bacterial metabolism can further be metabolized to various derivatives and absorbed into the human body [15].

Due to the great diversity of species forming the intestinal microbiota in different individuals, the profile of polyphenol metabolites that are generated and their final effect on the body are highly variable within the human population. The dietary polyphenols can be metabolized by various pathways leading to the formation of a number of different phenolic derivatives characterized by small and low molecular weight as well as a modified biological activity. For example, aglycones and oligomers are released by microbial glycosidases and esterases, which enhances their absorption [229]. On the other side, released aglycones can inhibit intestinal microbiota growth and activity, preventing the metabolism of other polyphenolic compounds from the diet. Some reactions of bacterial metabolism really improve the bioavailability and activity of polyphenolic compounds. In many situations, only the product of bacterial metabolism of a polyphenol can be absorbed and exert a beneficial impact in humans. However, other bacterial metabolites may be harmful to human cells or other members of the microbiota. Hence, apart from interindividual variation in a daily intake of polyphenols, interindividual differences in the composition of the human microbiota may lead to differences in bioavailability and bioefficiency of polyphenols and their metabolites and cause a different impact on host health [230,231,232].

The identified pathways of bacterial metabolism of the most important groups of polyphenolic compounds are presented below.

### 5.1. Isoflavonoids

One of the best examples of how significant the role is of the intestinal microbiota in polyphenols impact on human health are the nonsteroidal estrogens. A lack of particular species within the microbiota may cause that isoflavonoid cannot exert its expected effect even though it has been consumed. Isoflavonoids, including daidzein, genistein, and glycitein, are present in soybeans, but they are rather inactive. Only their metabolites, e.g., S-equol or O-desmethylangolensin (O-DMA), are able to exert their pro-healthy effects. Equol, because of its high binding affinity to the estrogen receptor (S-equol preferentially activates estrogen receptor ERβ), can alleviate the symptoms of menopause. Moreover, the antiandrogenic activity and inhibition of osteoclast formation, anticancer activities and anti-inflammatory effects have been observed [6]. It was demonstrated that equol has about 100 times higher estrogenic activity than the daidzein itself [232]. Although O-DMA did not exhibit agonistic or antagonistic activities toward the glucocorticoid receptor (TRa1, or TRb1) and has very weak agonistic activities against ERα and ERβ, it can influence the growth of cancer cells, osteoclast formation, scavenging superoxide radical or exert leptin secretion inhibitory activity [233]. Bacteria strains producing small to moderate amounts of dihydrodaidzein and/or O-DMA from daidzein and dihydrogenistein from genistein are recognized more often than equol producers [234]. O-DMA is found in 80–90% of the human population, whereas equol is found in only 30–50% of the population [235].

The possible metabolic pathways of daidzin and genistin degradation by bacteria are presented in Figure 4 and Figure 5, respectively.

The isoflavones biotransformation generally starts with glycoside hydrolysis to release the aglycon. For example, daidzin (daidzein 7-*O*-glycoside) can be hydrolyzed to daidzein by *Eubacterium ramulus* [244]. Then, the hydrogenation of the double bond between C2 and C3 in ring C of daidzein (DZN) and genistein (GN) generates dihydroisoflavones such as dihydrodaidzein (DHD) and dihydrogenistein (DHG), respectively [236]. Dihydroisoflavones are further subjected to bacterial metabolism and can undergo: (a) the reductive pathway leading to equol formation, (b) the cleavage of the C ring, followed by the fission of the molecules into two moieties. Equol is generally produced from daidzein through the reductive metabolism, through dihydrodaidzein (DHD), tetrahydrodaidzein (cis-THD and/or trans-THD) and dehydroequol (DE) as intermediates; however, some equol-producing bacteria have also been shown to convert the genistein into dihydrogenistein and finally to 5-hydroxy-equol [245,246]. *Clostridium* sp. strain HGH136 cleaved the C-ring of daidzein to produce O-desmethylangolensin, probably via 2-dehydro-*O*-demethylangolensin [241]. O-DMA may be further partially metabolized to resorcinol and 2-(4-hydroxyphenyl) propionic acid [233].

Daidzein was in part degraded by *E. ramulus* to *O*-desmethylangolensin, while genistein was completely degraded via 6′-hydroxy-*O*-desmethylangolensin to 2-(4-hydroxyphenyl)-propionic acid [243]. It means that the OH group in position 6′ of O-DMA was crucial for its further degradation. It is interesting that dihydrogenistein was neither observed as an intermediate in this transformation nor converted itself by growing cells of *E. ramulus*. Genistein-7-*O*-glucoside was partially transformed by way of genistein to the product 2-(4-hydroxyphenyl)-propionic acid.

*E. ramulus*, strain CG19-1 is capable of cleaving both 6′-hydroxy-*O*-desmethylangolensin and *O*-desmethylangolensin to phloroglucinol and resorcinol, respectively; and 2-(4-hydroxyphenyl) propionic acid was additionally formed from both O-DMA and 6′-OH-*O*-DMA [239].

A different metabolic pathway was revealed by Murota et al. [235]. They reported that the metabolites of genistein and glycitein that are primarily found in human urine were dihydrogenistein, 6ʹ-OH-*O*-DMA, 2-(4-hydroxyphenyl)-propionic acid and phloroglucinol for genistein, while dihydroglycitein, 5ʹ-methoxy-*O*-DMA and 6-methoxy-equol for glycitein. Moreover, strain CG19-1 cleaved both O-desmethylangolensin and 6′-hydroxy-*O*-desmethylangolensin to yield 2-(4-dihydroxyphenyl) propionic acid. The corresponding cleavage product, resorcinol, was only observed for O-desmethylangolensin.

According to Rossi et al. [242], the metabolites arising from glycitein include dihydroglycitein, which can be further O-demethylated to 6,7,4′-trihydroxyisoflavone (proved in vitro for *Eubacterium limosum*) and reduced to dihydro-6,7,4′-trihydroxyisoflavone, and further reduced to 6-hydroxyequol or cleaved to 5′-hydroxy-*O*-desmethylangolensin. The other pathway of dihydroglycitein degradation was through the C-ring cleavage producing 5′-*O*-methoxy-*O*-desmethylangolensin or reduction to 6-methoxy-equol (Figure 6).

*Slackia isoflavoniconvertens* is capable of contributing to the bioactivation of daidzein and genistein by the formation of equol and 5-hydroxy-equol, respectively [246].

It should be underlined that some bacteria can produce equol from either daidzein or its glycoside daidzin, but some cannot produce equol unless several other species of bacteria metabolize daidzin to aglycone daidzein and daidzein to DHD or other derivatives that are also present [235]. For example, *Clostridium* sp. strain HGH6 and *Lactobacillus* sp. Niu-O16 can reduce daidzein to dihydrodaidzein but did not convert dihydrodaidzein to equol [247]. On the other hand, *Eggerthella* sp. Julong 732 is capable of converting dihydrodaidzein, but not daidzein, to equol [238,247]. *Eggerthella* sp. Strain YY7918 converted substrates daidzein and dihydrodaidzein into S-equol but did not convert daidzin, glycitein, genistein, or formononetin into it [248]. Strain TM-40 (93% of homology with *Coprobacillus catenaformis*) isolated by Tamura et al. [249] produced dihydrodaidzein from both daidzein and daidzin. Decroos et al. [240] isolated from human feces a stable mixed microbial culture (*Enterococcus faecium* strain EPI1, *Lactobacillus mucosae* strain EPI2, *Finegoldia magna* strain EPI3 and an as yet undescribed species related to *Veillonella* sp.) that was able to covert daidzein into equol.

Among intestinal bacteria that were proved to metabolize the soya isoflavone daidzein and genistein to equol, DHD and/or O-DMA are *Slackia equolifaciens* (DZN to equol) [250], *Slackia isoflavoniconvertens*, *Adlercreutzia equolifaciens*, *Asaccharobacter celatus*, *Enterorhabdus mucosicola* (DZN to equol), *Peptoniphilus gorbachii* (DZN and GN to equol and O-DMA), *Gordonibacter urolithinfaciens* (DZN to O-DMA), some strains of *Eggerthella lenta* (DZN and GN to O-DMA), *Enterococcus lactis* (to O-DMA), some strains of *Bifidobacterium adolescentis* (DZN and GN to O-DMA), *B. animalis* (DZN to O-DMA) and *B. longum* (DZN and GN to O-DMA), some members of Coriobacteriaceae, e.g., *Collinsella massiliensis* (DZN and GN to O-DMA) and *C. aerofaciens* (DZN to O-DMA) [234], *Eggertella* strain Julong 732 (DHD via THD to equol) [238], *Lactococcus garvieae* strain 20–92 [251], *Eubacterium ramulus* Julong 601 (DZN to O-DMA, GN to 2-(4-hydroxyphenyl) propionic acid) [252], *Clostridium* sp. HGH 136 (DZN to DHD) [241] and HGH6 (DZN do DHD, GN to DHG), and *E. coli* HGH21 (DZN to DHD and GN to DHG) [253].

Puerarin is a daidzein 8-C-glucoside and was reported to be metabolized to daidzein by human intestinal flora such as *E. ramulus* CG 19-1 [239] or intestinal strain PUE, converting puerarin to daidzein by cleaving a C-glucosyl bond [254]. Formononetin and biochanin A are the principal isoflavones of red clover (and as a consequence, equol is present in cow milk) and can be consumed in the form of dietary supplements. Hur et al. [255] demonstrated that *Eubacterium limosum* is able to produce daidzein and genistein from formononetin and biochanin A, respectively. It means that due to bacterial metabolism, more potent phytoestrogens have been formed in the colon, as the estrogenic potencies of the mentioned compound for both estrogen receptors ERα and ERβ showed the affinities in the order of genistein > daidzein > biochanin A > formononetin. In the urine samples of volunteers consuming formononetin and biochanin A, other metabolites were also identified, such as dihydroformononetin and angolensin for formononetin and dihydrobiochanin A and 6′-hydroxyangolensin for biochanin A [256] (Figure 7).

### 5.2. Other Phytoestrogens

In addition to soy isoflavonoids, there are other ligands for estrogen receptors that are produced by intestinal microbiota, such as enterolactone, enterodiol, urolithins and 8-prenylnaringenin. Enterolactone and enterodiol are derivatives of plant lignans from sesame seed or flaxseed. It was proved at concentrations that can be achieved with high consumption of products rich in lignans, both, but enterolactone to a lesser extent can potently activate human estrogen receptors ERα and ERβ [257]. The bacterial transformation of lignans into phytoestrogens (Figure 8) was carried out mainly by *Peptostreptococcus* and *Eubacterium* species and included their demethylation and dihydroxylation, leading to enterolactone production [258]. Enterolactone can further be converted into enterodiol, and various studies proved that both the mentioned mammalian lignans are produced by human colonic microbiota from dietary precursors. Production of enterodiol is about 2000 times more efficient, meaning that the enterodiol-producing bacteria are dominant in the human gut.

The main bacteria converting lignans to enterolactone and enterodiol are *Peptostreptococcus* sp. SDG-1 and *Eubacterium* sp. SDG-2 [262], *Bacteroides distasonis*, *B. fragilis*, *B. ovatus*, *Eubacterium callanderi*, *Eubacterium limosum*, *Clostridium cocleatum*, *Clostridium scindens*, *Eggerthella lenta*, *Butyribacterium methylotrophicum*, *Butyribacterium pseudocatenulatum*, *Bifidobacterium longum*, *B* > *breve*, *B. catenulatum*, *B. pseudocateunaltum*, *Enterococcus faecalis*, *Ruminococcus* sp. END-1 [261], *Clostridium saccharogumia* and *Lactonifactor longoviformis* [263]. Two organisms able to demethylate and dehydroxylate secoisolariciresinol were isolated from human feces. Based on 16S rRNA gene sequence analyses, they were named *Peptostreptococcus productus* SECO-Mt75m3 and *Eggerthella lenta* SECO-Mt75m2 [264]. It was demonstrated both in vivo and in vitro that the major metabolite of sesamin in humans is enterolactone [265]. The intestinal pathways of enterolactone and enterodiol production from lignan are presented in Figure 8.

Ellagitannins (ELT) are one of the main groups of hydrolyzable tannins that are characterized by high antioxidant activity. They are common in some fruits, such as pomegranates, black raspberries, raspberries and strawberries, as well as in walnuts and almonds. Chemically they are different esters of hexahydroxydiphenic acid (HHDP) and a polyol, usually glucose or quinic acid [266]. According to the number of HHDP groups linked to the sugar moiety, ellagitannins can be classified into monomeric, oligomeric, and polymeric ELT. The main ellagitannins identified in foods are punicalagin (Figure 9), sanguiin H-6 (dimer of casuarictin) **(212)**, lambertianin C (trimer of casuarictin) **(214)**, pedunculagin, castalagin **(209)**, casuarictin **(210)** and potentillin. Because of their size (634 Da for sanguiin H4 to up to 3740 Da for lambertianin D), these molecules are characterized by very low bioavailability and are not absorbed in the gastrointestinal tract until they are metabolized by gut bacteria. Intact ellagitannins and a product of their acidic or basic hydrolysis—ellagic acid (Figure 9), reach the distal part of the gastrointestinal tract where they are transformed by intestinal microbiota into dibenzopyran-6-one derivatives, known as urolithins, that are much better absorbed [267].

The generic name of urolithins includes different hydroxylated 6H-dibenzo[b,d] pyran-6-one derivatives. The bacterial transformation includes reduction of one of the two lactone groups followed by decarboxylation and sequential dehydroxylation involving a step-by-step reduction to tetrahydroxy (urolithin D), trihydroxy (urolithin C), dihydroxy (urolithin A and isourolithin A), and monohydroxy dibenzopyranones (urolithin B). The pathway of bacterial metabolism is presented in Figure 10. Bacteria able to catalyze the biotransformation of ellagitannins to urolithins are *Gordonibacter urolithinfaciens* and *G. pamelaeae* that belong to the family Coriobacteriaceae [269,270] and *Ellagibacter isourolithinifaciens* from Eggerthellaceae [271].

Although urolithins are characterized by lower antioxidant activity than ellagitannins, they circulate in the plasma as glucuronide and sulfate conjugate and display benefit influence on human health due to their estrogenic and/or anti-estrogenic activity, as well as anticancer activities. It means that bacterial metabolism is crucial for the pro-healthy properties of various berries [266,272].

It has been shown that the production of the potent hop phytoestrogen 8-prenylnaringenin (8-PN) depends on the activity of human intestinal microbiota [275]. This compound is generated from xanthohumol and isoxanthohumol that unaltered reach the small intestine (Figure 11). Among bacteria that catalyze the demethylation of isoxanthohumol into 8-PN are *Eubacterium limosum* and *E. ramulus* [275,276,277]. In addition to a strong impact on the ERα receptor, 8-prenylnaringenin inhibits angiogenesis and metastasis, prevents bone loss in rats and exhibits antiandrogenic activity [275,278].

### 5.3. Bacterial Transformation of Anthocyanidins

Anthocyanidins (ACD) are plant pigments responsible for flower, fruit and vegetable color. Their structure and color depend on pH value and the presence of copigments. In plant tissues, ACD are generally present in the form of glycosides, called anthocyanins (ACN), that are susceptible to hydrolytic conversion into their corresponding anthocyanidins. Glucose, galactose, rhamnose and arabinose are the sugars most commonly encountered, usually as 3-*O*-glycosides or 3,5-*O*-diglycosides; however, rutinosides (6-*O*-L-rhamnosyl-D-glucosides), sophorosides (2-*O*-D-glucosyl-D-glucosides) and sambubiosides (2-*O*-D-xylosyl-D-glucosides) also occur, as do some 3,7-diglycosides and 3-triosides [280]. Moreover, some of the hydroxyl groups can be methylated, giving the big diversity of plant anthocyanidins. The anthocyanidins occur in the vacuole as an equilibrium of four molecular species that affects their color (Figure 12). However, after fruit and vegetable consumption, the form of the flavylium cation exists only in the stomach, while other forms are present in the lower parts of the gastrointestinal tract and in the tissues (if absorbed).

Bacterial metabolism of ACN involves the cleavage of glycosidic linkage and breakdown of the anthocyanidin heterocycle. Aura et al. [281] demonstrated that cyanidin-3-rutinoside was degraded through cyanidin-3-glucoside and cyanidin aglycone as intermediary metabolites. After hydrolysis of the protective 3-glycosidic linkage, the released aglycons are stable under acidic pH but unstable under neutral or slightly basic pH. It means that under physiological conditions in the small intestine, the cleavage of the heterocyclic flavylium ring occurs [274]. An attack of the flavylium carbon at position 2 produces an unstable hemiketal that rapidly forms a ketone (Figure 13). Through keto-enol tautomerism of the neighboring enol functionality, the resulting α-diketone is very reactive and is easily decomposed by gut microbiota to phenolic acids (mainly protocatechuic acid, syringic acid, vanillic acid) and aldehydes (mainly phloroglucinol aldehyde) [274].

It was demonstrated for raspberry anthocyanins; when incubated with fecal suspensions under anaerobic conditions, that they underwent a transformation by the colonic microflora. After C-ring fission in cyanidin, aglycone phenolic acids were released, originating from both the A and B rings. It was proved that some of the colonic catabolites entered the circulation and were further metabolized before being excreted in urine (e.g., as hippuric acid) [283].

Phenolic acids may be utilized as a source of energy by the intestinal microflora. Keppler and Humpf [282] demonstrated that bacterial metabolism of the methoxyl derivatives as syringic acid and vanillic acid was accompanied by O-demethylation and resulted in the formation of gallic acid and protocatechuic acid (PCA), respectively. As phloroglucinol aldehyde (PHA) was degraded by the intestinal microflora very similar in comparison to the sterilized control samples, it was not possible to distinguish between the chemical or microbial transformation of the aldehyde. However, the phloroglucinol acid was detected as the oxidation product of PHA in very low amounts only in the non-sterilized inoculum filtrate [282], indicating the role of gut microbiota in the transformation.

Some in vitro studies revealed that the numbers of potentially beneficial bacteria (bifidobacteria and lactobacilli) increased after the consumption of purple sweet potato anthocyanins and grape seed extract [284]. As anthocyanins are hardly absorbed in the small intestine, they may be transformed into small molecular phenolic acids by colonic microbiota through ring cleavage, dihydroxylation and methylation reactions. Such metabolites generated from polyphenols may selectively stimulate the growth of beneficial bacterial, whereas the proliferation of harmful bacteria would be inhibited. Ávila et al. [285] analyzed various strains of *Lactobacillus plantarum* and *L. casei*, as well as probiotic strains *Lactobacillus acidophilus* LA-5 and *Bifidobacterium lactis* BB-12. They proved the enzymatic potential of selected strains for bioconversion of delphinidin and malvidin glycosides to their metabolites. Incubation of malvidin-3-glucoside with *B. lactis* BB-12, *L. plantarum* IFPL722, and *L. casei* LC-01 cell-free extracts led to different patterns of gallic, homogentisic and syringic acid formation.

It was also reported that gallic acid and free anthocyanins activated cell growth and the rate of malic acid degradation; vanillic acid showed a slight inhibiting effect, while protocatechuic acid had no effect. Finally, gallic acid and ACN were metabolized, especially by growing cells [88]. Incubation of malvidin-3-glucoside with fecal bacteria mainly resulted in the formation of syringic acid, while the mixture of anthocyanins resulted in the formation of gallic, syringic and *p*-coumaric acids [286].

The most abundant anthocyanins in fruit and vegetables are cyanidin, pelargonidin, petunidin, peonidin and delphinidin. The hypothesized pathways of their bacterial degradation are presented in Figure 14 and Figure 15. Mayor ACN metabolites generated in the human colon by bacteria are protocatechuic acid, syringic acid, vanillic acid, gallic acid, phenylacetic acid, 3,4-dihydroxyphenylpropionic acid, 3,4-dihydroxyphenylacetic acid, 4-hydroxybenzoic acid, but also 4-hydroxyphenylethanol (tyrosol), catechol, benzoic acid [282,284,287,288,289].

Zhu et al. [284] reported that 2,4,6-trihydroxybenzoic acid, 4-hydroxybenzaldehyde, benzoic acid, phenylacetic acid, and phenylpropionic acid were found in the medium after bacterial metabolism od cyanidin-3-*O*-glucoside. The metabolism of cyanidin-3-*O*-glucoside and cyanidin-3-*O*-rutinoside mainly resulted in the formation of protocatechuic, vanillic, and *p*-coumaric acids, as well as 2,4,6-trihydroxybenzaldehyde, while the main metabolites of delphinidin-3-*O*-rutinoside were gallic acid, syringic acid and 2,4,6-trihydroxybenzaldehyde. Among minor metabolites identified after microbial metabolism of mentioned glycosides were: protocatechuic acid-glucoside, caffeic acid, tartaric acid, catechol, as well as pyrogallol, ferulic acid, 4-hydroxybenzoic acid [290]. This research indicated that the intake of ACNs might result in the appearance of specific metabolites that exert a protective effect in host physiology.

The main phenolic acid detected in fecal suspensions incubated with raspberry anthocyanins were: 3-phenylacetic acid, 3-(4′-hydroxyphenyl) lactic acid, tyrosol, 3-(4′-hydroxyphenyl) propionic acid, 3-(3′-hydroxyphenyl) propionic acid, 4-hydroxybenzoic acid and 3,4-dihydroxybenzoic acid, but lower amounts of catechol, resorcinol, pyrogallol and 3-(3′,4′-dihydroxyphenyl) propionic acid were also found [283]. Seven metabolites formed by human fecal bacteria were observed by LC/MS after incubation with cyanidin-3-*O*-glucoside and cyanidin-3-*O*-galactoside, and protocatechuic acid (major metabolite), 2,4,5-trihydroxy-benzaldehyde, 5-hydroxy-2-(4′-hydroxyphenyl)-2H-chromen-3,7-dione, 3,5,7-trihydroxy-2-(3′,4′-dihydroxyphenyl)-2H-chromene, and 5-hydroxy-2-phenyl-2H-chromen-3,7-dione were identified [288]. The deglycosylation, decomposition, hydrogenation, and dihydroxylation reactions were involved in their generation.

Malvidin-3-glucoside was completely degraded into syringic acid after incubation with a human fecal slurry for 24 h, whereas gallic acid, *p*-coumaric, and syringic acid were formed after a mixture of various anthocyanins were incubated with healthy human fecal bacteria [291].

### 5.4. Metabolism of Procyanidins and Catechins by Intestinal Bacteria

Condensed tannins (also called catechol-type tannins or non-hydrolyzable tannins) are an important class of polyphenols that do not contain sugar residues. They are also called proanthocyanidins as, under oxidative conditions, they depolymerize, yielding anthocyanidins. Therefore, different types of condensed tannins exist, such as the procyanidins, propelargonidins, prodelphinidins, profisetinidins, proteracacinidins, proguibourtinidins or prorobinetidins.

Condensed tannins are formed from flavan-3-ols or flavan-4-ols. These particular types of condensed tannins are procyanidins, which are not susceptible to cleavage by hydrolysis. Procyanidins are polymers of 2 to 50 (or more) catechin units (usually catechin and epicatechin molecules) joined by carbon–carbon bonds. The most ubiquitous are B-type procyanidins, abundant in apple, cocoa, pear, blueberries; these subunits are linked by single bond C4–C8 or C4–C6. In A-type procyanidins, present in cranberries, a double linkage exists; the C4–C8 or C4-C6 bond is accompanied by an additional C2–O–C7 or C2–O–C5 ether bond.

Some studies demonstrated that highly polymeric procyanidins (PPs) administration markedly decreased the Firmicutes/Bacteroidetes ratio and increased by eight times the proportion of *Akkermansia*, suggesting that PPs influence the gut microbiota and the intestinal metabolome to produce beneficial effects on metabolic homeostasis [76]. On the other hand, some species of intestinal bacteria are able to degrade oligomeric procyanidins. Spencer et al. [221] proved that procyanidin oligomers (trimer to hexamer) are hydrolyzed in simulated gastric juice to mixtures of epicatechin monomer and dimer, thus enhancing the potential for their absorption in the small intestine. Proanthocyanidins that undergo partial acid-catalyzed cleavage then decompose to monomeric flavan-3-ols, which are also metabolized by colonic bacteria. The bacterial degradation of flavan-3-ols and proanthocyanidins follow a similar pathway, and both lead to a generation of a unique compound 5-(3′,4′-dihydroxyphenyl)-γ-valerolactone, which further undergoes dihydroxylation and oxidation to produce phenolic acids [291,293] (Figure 16).

The incubation of purified (+)-catechin, (−)-epicatechin, procyanidins A2 and B2, as well as partially purified apple and cranberry procyanidins with human gut microbiota, resulted in their degradation. The common metabolites were benzoic acid, 2-phenylacetic acid, 3-phenylpropionic acid, 2-(3′-hydroxyphenyl) acetic acid (OPAC), 2-(4′-hydroxyphenyl) acetic acid, 3-(3′-hydroxyphenyl) propionic acid, and hydroxyphenylvaleric acid. Interesting, that 5-(3′,4′-dihydroxyphenyl)-γ-valerolactone and 5-(3′-hydroxyphenyl)-γ-valerolactone were identified as the bacterial metabolites of epicatechin, catechin, procyanidin B2, and purified apple procyanidins, but not from the procyanidin A2 or cranberry procyanidin ferments, while 2-(3′,4′-dihydroxyphenyl) acetic acid was only found in the fermented broth of procyanidin B2, A2, apple, and cranberry procyanidins [298].

The monomeric flavan-3-ols, which are usually shared by condensed tannins, can be degraded by fecal microbiota into low molecular weight aromatic compounds, including phenylpropionic acid, 2-(3′-hydroxyphenyl) acetic acid, 3-(3′-hydroxyphenyl) propionic acid, 5-(3′-hydroxyphenyl) valeric acid, phenylacetic acid, and 2-(4′-hydroxyphenyl) acetic acid [294]. Similar results have been obtained Appeldoorn et al. [296]. Purified procyanidin dimers, when incubated with human microbiota, have been transformed and among major identified metabolites were 2-(3′,4′-dihydroxyphenyl) acetic acid (DOPAC) and 5-(3′,4′-dihydroxyphenyl)-γ-valerolactone. Other metabolites detected were: OPAC, 2-(4′-hydroxyphenyl) acetic acid, 3-(3′-hydroxyphenyl) propionic acid, phenylvaleric acids, monohydroxylated phenylvalerolactone, and 1-(3′,4′-dihydroxyphenyl)-3-(2′′,4′′,6′′-trihydroxyphenyl) propan-2-ol. In studies of Deprez et al. [297], polymeric procyanidins were metabolized by colonic bacteria into low-molecular-weight phenolic acid, and the main metabolites were 3-phenylpropionic acid, 3-(4′-hydroxyphenyl) propionic acid, 3-(3′-hydroxyphenyl) propionic acid, 5-(3′-hydroxyphenyl) valeric acid, 2-(3′-hydroxyphenyl) acetic acid, and 2-(4′-hydroxyphenyl) acetic acid.

Among intestinal bacteria that are able to convert catechins are *Eggerthella lenta* and *Flavonifractor plautii* (formerly *Clostridium orbiscindens*) [295]. *Eggerthella lenta* rK3 reductively cleaved the heterocyclic C-ring of both (−)-epicatechin and (+)-catechin giving rise to 1-(3′,4′-dihydroxyphenyl)-3-(2′′,4′′,6′′-trihydroxyphenyl) propan-2-ol (Figure 16). The conversion of catechin proceeded five times faster than that of epicatechin. *Flavonifractor plautii* aK2 and *Flavonifractor plautii* DSM 6740 further converted 1-(3′,4′-dihydroxyphenyl)-3-(2′′,4′′,6′′-trihydroxyphenyl) propan-2-ol to 5-(3′,4′-dihydroxyphenyl)-γ-valerolactone and 4-hydroxy-5-(3′,4′-dihydroxyphenyl) valeric acid.

According to Tzounis et al. [59], the initial conversion of (+)-catechin to (+)-epicatechin is required to the generation of 5-(3′,4′-dihydroxyphenyl)-γ-valerolactone, 5-phenyl-γ-valerolactone and phenylpropionic acid as metabolites. The prebiotic effects of both (+)-catechin and (−)-epicatechin was observed, suggesting that the consumption of flavanol-rich foods may support gut health through their ability to exert prebiotic actions.

Monagas et al. [299] demonstrated that some phenolic acids, including 3-*O*-methyl gallic, gallic, caffeic, 3-(4′-hydroxyphenyl) propionic, phenylpropionic, and 2-(4′-hydroxyphenyl) acetic acids derived from the microbial degradation of tea catechins, were able to inhibit the growth of several pathogenic and non-beneficial intestinal bacteria without significantly affecting the growth of beneficial bacteria (*Lactobacillus* spp. and *Bifidobacterium* spp.). It is possible that *Bifidobacterium* sp. are resistant to flavan-3-ols, being the are important iron-chelating compounds, because these bacteria do not use heme-containing enzymes [237]. Growth of certain pathogenic bacteria such as *Clostridium perfringens*, *Clostridium difficile*, *Streptococcus pyogenes*, and *Str. pneumoniae* was significantly repressed by tea phenolics (catechin, epicatechin, gallic acid, caffeic acid), while commensal anaerobes like *Clostridium* spp., *Bifidobacterium* spp. and probiotics such as *Lactobacillus* sp. were less severely affected [70]. Similarly, the bacterial metabolites, such as 3-(4′-hydroxyphenyl) propionic acid, 3-phenylpropionic acid and 2-(4′-hydroxyphenyl) acetic acid, strongly inhibited the growth of *E. coli, S. aureus* and *Salmonella* sp. without influencing beneficial *L. casei* strain Shirota and *Bifidobacterium breve*.

Alakomi et al. [300] have shown that DOPAC, OPAC, 3-(3′,4′-dihydroxyphenyl) propionic acid, 3-(4′-hydroxyphenyl) propionic acid, 3-phenylpropionic acid, and 3-(3′-hydroxyphenyl) propionic acid efficiently destabilized the outer membrane of *Salmonella enterica* subsp. *enterica* serovar Typhimurium and *S. enterica* subsp. *enterica* serovar Infantis. Moreover, DOPAC, OPAC and 3-(3′,4′-dihydroxyphenyl) propionic acid increased the susceptibility of *Salmonella* Typhimurium strains for novobiocin. It means that beneficial bacteria residing in the human gut can inhibit *Salmonella* growth by the transformation of food flavonoids to active antimicrobial metabolites.

### 5.5. The Bacterial Metabolism of Flavones and Flavonols

It is interesting that some compounds are common and can be generated by colonic microbiota during the metabolism of various polyphenols, although to a different extent. For example, flavan-3-ols, as well as flavonols and hydroxycinnamic acids, lead to the generation of 3-(3′,4′-dihydroxyphenyl)-propionic acid, 3-(3′-hydroxyphenyl) propionic acid, and 3-(4′-hydroxyphenyl) propionic acid. It means that some enzymes and metabolic pathways are quite common among bacteria. When quercetin is metabolized by bacteria, ring fission is done, leading to the generation of DOPAC, OPAC and protocatechuic acid (PCA) (Figure 17). Braune et al. [301] examined the degradation mechanism of the flavonol quercetin and the flavone luteolin and had demonstrated that *Eubacterium ramulus* converted quercetin through taxifolin and alphitonin, resulting in the formation of DOPAC and phloroglucinol. Flavonol luteolin was transformed by *E. ramulus* to 3-(3′,4′-dihydroxyphenyl) propionic acid via eriodictyol and derivatives. In both pathways, ring fission had taken place. Glycosides of quercetin, such as common in onions quercetin 4′-*O*-glucoside and quercetin 3-*O*-glucoside, are first hydrolyzed to aglycone and then are also catabolized, giving ring-fission products [235]. *Bifidobacterium animalis* subsp. *lactis* AD011, isolated from infant feces, has been shown to catalyze quercetin 3-*O*-glucoside and isorhamnetin 3-*O*-glucoside into quercetin and isorhamnetin, respectively [302].

The degradation of flavones and flavonols by *Clostridium orbiscindenss* was studied by Schoefer et al. [303]. They confirmed the quercetin degradation via taxifolin and alphitonin. Flavone apigenin and luteolin were converted to 3-(4′-hydroxyphenyl) propionic acid and 3-(3′,4′-dihydroxyphenyl) propionic acid, respectively, and phloroglucinol was released in both cases (Figure 17). The intermediate metabolites were naringenin and phloretin for apigenin and eriodictyol and dihydrochalcone for luteolin [303]. However, the isolated *C. orbiscindens* strain was unable to hydrolyze the glycosidic bonds of luteolin 3-*O*-glucoside, luteolin 5-*O*-glucoside, naringenin 7-*O*-neohesperidoside (naringin), quercetin 3-*O*-glucoside, quercetin 3-*O*-rutinoside (rutin), and phloretin 2′-*O*-glucoside, suggesting that other bacteria are required for the initial steps in the metabolism of flavonoid glycosides in the human intestine. Similar pathways were reported for degradation of myricetin, kaempferol as well as quercetin, apigenin and luteolin glycosides, and among bacteria involved in their metabolism were *Enterococcus casseliflavus*, *Eubacterium ramulus*, *Eubacterium oxidoreducens*, *Butyrivibrio* spp., *Clostridium orbiscidens*, *Eggerthella* sp., *Flavonifractor plautii*, *Bacteroides uniformis*, *Bacteroides ovatus*, *Bifidobacterium* spp., *Bacteroides distasonis*, and *Blautia* sp. [235,237,304,305,306,307].

Experiments using radiolabeled quercetin 4′-*O*-glucoside (Q4ʹG) revealed that Q4ʹG passes through the gastrointestinal tract of rats and that almost all of Q4ʹG is converted into phenolic acids, with DOPAC and OPAC being the most abundant, and a small amount of PCA was also generated [235]. Moreover, 69% of Q4′G radioactivity was recovered in the form of phenolic acid derivatives, such as OPAC and hippuric acid, in the urine. It means that the first ring-fission product is DOPAC, which is subsequently subjected to dehydroxylation to form OPAC, followed by further catabolism into hippuric or benzoic acids (Figure 17). DOPAC also has been identified as a major catabolite of quercetin glycosides, such as rutin, as well as procyanidins (Figure 16). It is important because DOPAC is known to be a metabolite of the neurotransmitter dopamine, suggesting the existence of a metabolic pathway for DOPAC in humans. It has been demonstrated that DOPAC exerts anticancer, anti-inflammatory, cardioprotective and neuroprotective impact. However, DOPAC may inhibit mitochondrial respiration in brain mitochondria (when NO radical is present) and thus lead to mitochondrial dysfunction, which is assumed to be an important mechanism involved in Parkinson’s disease [308].

### 5.6. Microbial Catabolism of Phenolic Acids

Phenolic acids can be delivered to the intestine with food, but they are also generated as the final metabolites during the degradation of various polyphenols. Phenolic acids play an important protective role in degenerative diseases as they exert antioxidant, antitumor, apoptotic, neuroprotective, hepatoprotective, anti-inflammatory and antimicrobial properties [35]. However, there has been some controversy about the bioactivity of polyphenols after metabolism. Once ingested, these molecules are metabolized and transformed into methylated, glucuronated and sulfated metabolites, and there is much evidence proving both the enhanced and decreased biological activity of phenolic acid metabolites [35]. Not all phenolic acids are absorbed, and some of them reach the colon and can be metabolized by bacteria. It is supposed that the presence of an ester moiety lowers hydroxycinnamic acids (HCAs) absorption. Actually, HCAs in a free form are rapidly absorbed throughout the gastrointestinal tract, while HCAs esters or HCAs attached to cell walls require to be hydrolyzed by bacterial esterases before absorption [311]. As a large interindividual variation of phenolic acid metabolites (Figure 18) was observed, it may suggest that the catabolic pathways of both chlorogenic acid and other phenolic acids depend mainly on the colon microbiota composition.

Incubation of coffee samples (a rich source of phenolic acids) with the human fecal microbiota led to the rapid metabolism of chlorogenic acid and the production of dihydrocaffeic acid and dihydroferulic acid, while caffeine remained unmetabolized [316]. Caffeic acid esters can be rapidly transformed to 3-(3′-hydroxyphenyl) propionic acid by human fecal microbiota [304] by de-esterification followed by a reduction of a double bond and dehydroxylation at the C4 position (Figure 18). Monteiro et al. [317] revealed that the main chlorogenic acid metabolites identified in urine after coffee consumption were: dihydrocaffeic, gallic, isoferulic, ferulic, vanillic, caffeic, 5-*O*-caffeoylquinic, sinapic, 4-hydroxybenzoic, and *p*-coumaric acids, with gallic and dihydrocaffeic acids being the major ones. Similar results were reported by Clifford et al. [37]. One of the most abundant sources of caffeic acid in nature is 5-*O*-caffeoylquinic acid (neochlorogenic acid), which was also proved to be hydrolyzed to caffeic and quinic acids by esterases from colonic microflora and is not degraded and absorbed in the upper gastrointestinal tract [311]. The lack of colonic microbiota (e.g., in germfree rats) resulted in the inhibition of hippuric acid formation, indicating that esterase enzymes of the colonic microbiota are involved in this pathway [318].

### 5.7. Bacterial Metabolism of Resveratrol and Curcumin

Resveratrol is a natural polyphenol widely found in its *trans* isomer form in various fruits, especially grapes and berries, peanuts, and red wine. It was reported that purified resveratrol inhibited the growth of some pathogens, among other intestinal bacteria such as *Helicobacter pylori*, *Enterococcus faecalis*, *Pseudomonas aeruginosa*, *Vibrio* spp. [319]. However, some bacteria are able to metabolize *trans*-resveratrol. *Slackia equolifaciens* and *Adlercreutzia equolifaciens* [320] and *Eggerthella lenta* ATCC 4305 [319] converted resveratrol to dihydroresveratrol (Figure 19); while *Bacillus cereus* NCTR-466, *Achromobacter denitrificans* NCTR-774, and *E. coli* ATCC 47,004 metabolizes trans-resveratrol into resveratrol 3-*O*-glucoside (piceid) and resveratrol 4-*O*-glucoside (resveratroloside) [319]. Among other colonic metabolites of resveratrol, lunularin and 3,4′-dihydroxy-*trans*-stilbene [320] were identified. However, their bacterial producers are unknown (Figure 19). The 16S rRNA sequencing of fecal samples demonstrated the association of lunularin producers with a higher abundance of Bacteroidetes, actinobacteria, Verrucomicrobia, and Cyanobacteria and with a lower abundance of Firmicutes than either the dihydroresveratrol or mixed producers [321]. The bacterial metabolites of resveratrol can exert a beneficial impact on human health. Dihydroresveratrol reduced fatty acid-binding protein-4 expression, involved in fatty acid uptake in human macrophages treated with oxidized LDL and stimulates fatty acid oxidation in human fibroblasts, lunularin reduced the expression of proinflammatory mediators in endothelial cells [322], while 3,4′-dihydroxy-*trans*-stilbene increased glucose uptake and induced adenosine monophosphate kinase phosphorylation in C2C12 myotubes independently of insulin [323].

Jarosova et al. [324] examined the metabolism of six stilbenoids resveratrol, oxyresveratrol, piceatannol, thunalbene, batatasin III, and pinostilbene by colon microbiota from various donors. It was demonstrated that resveratrol, oxyresveratrol, piceatannol and thunalbene were subjected to metabolic transformation via double bond reduction, dihydroxylation, and demethylation (Figure 19), while batatasin III and pinostilbene were stable at simulated colon conditions. Authors reported strong interindividual differences in speed, intensity, and pathways of metabolism among the fecal samples obtained from the donors, suggesting that microbiota composition plays a crucial role in the influence of resveratrol on human health.

Curcumin is a lipophilic polyphenol characterized by quite poor bioavailability. It is supposed that curcumin passes through the stomach without any chemical modifications and reaches the large intestine, where it undergoes extensive phase I and II metabolism. The reductive pathways of metabolism by phase I enzymes lead to the formation of dihydrocurcumin, tetrahydrocurcumin, and hexahydrocurcumin (Figure 20) [325].

However, consecutive reduction of the double bonds in the curcumin chain resulting in the formation of dihydrocurcumin, tetrahydrocurcumin, and hexahydrocurcumin can occur in the gut by a CurA reductase (NADPH-dependent curcumin/dihydrocurcumin reductase) that has been isolated from intestinal *E. coli* [326]. The 24-h fermentation of curcumin, demethoxycurcumin and bis-demethoxycurcumin by human fecal microbiota resulted in 24%, 61% and 87% degradation, respectively. Three main metabolites were identified: tetrahydrocurcumin, dihydroferulic acid and 1-(4-hydroxy-3-methoxyphenyl)-2-propanol [327]. A similar experiment was performed by Burapan et al. [328], but a mixture composed of curcumin, demethoxycurcumin, and bis-demethoxycurcumin was metabolized by the human intestinal bacterium *Blautia* sp. MRG-PMF1. New metabolites generated from curcumin and demethoxycurcumin by the methyl aryl ether cleavage reaction were identified. Demethylcurcumin and bisdemethylcurcumin were sequentially produced from curcumin, while demethyldemethoxycurcumin was produced from demethoxycurcumin [328]. Bis(demethyl)tetrahydrocurcumin and bis(demethyl)-hexahydrocurcumin were identified among colonic metabolites of curcumin, demethoxycurcumin and bis-demethoxycurcumin [329].

All these metabolites can undergo phase II metabolism by glucuronidases and sulfotransferases that are capable of conjugating glucuronic acid or sulfate molecule, respectively, to produce the corresponding glucuronide and sulfate O-conjugated metabolites. Furthermore, gut microbiota may deconjugate the phase II metabolites and convert them back to the corresponding phase I metabolites or to fission products such as ferulic acid and dihydroferulic acid in the colon [325].

It is interesting that CurA reductase, besides the ability to conversion of curcumin to tetrahydrocurcumin, is also able to metabolize resveratrol [326].

## 6. Conclusions

This review describes a bidirectional relationship between polyphenols delivered with food and the human gut microbiota. The manuscript presents a compilation of the knowledge from two perspectives. The first part describes the impact of various polyphenols classes on bacteria, with particular emphasis on human intestinal microbiota representatives. The mechanism of inhibitory impact of polyphenols, including protein binding, inhibition of nucleic acid synthesis, interaction with the cell wall and bacterial membranes, substrate deprivation, inhibition of energy metabolism and changes in cell attachment and biofilm formation, are discussed in details. In the second part, the role and pathways of bacterial biotransformation of polyphenols are described, especially those reactions where bioactive metabolites with a significant impact on the human organism (both positive and negative) are produced. The role of interindividual variation in microbiota composition in the impact of food polyphenols on human health is explained. For example, the biotransformation of isoflavonoids and other phytoestrogens to bioactive O-DMA and S-equol, the generation of urolithins, the bacterial metabolites that can cross the blood–brain barrier, the degradation of complex condensed tannins and lignans as well as catabolic pathways of low-molecular-weight phenolic acids are elucidated.

The exact structures of all discussed phenolic compounds can be found in tables and figures, which will facilitate their comparison with the structures of other food ingredients and drawing one’s own conclusions about their potential activity.

## Figures and Tables

**Figure 1 antioxidants-10-00188-f001:**
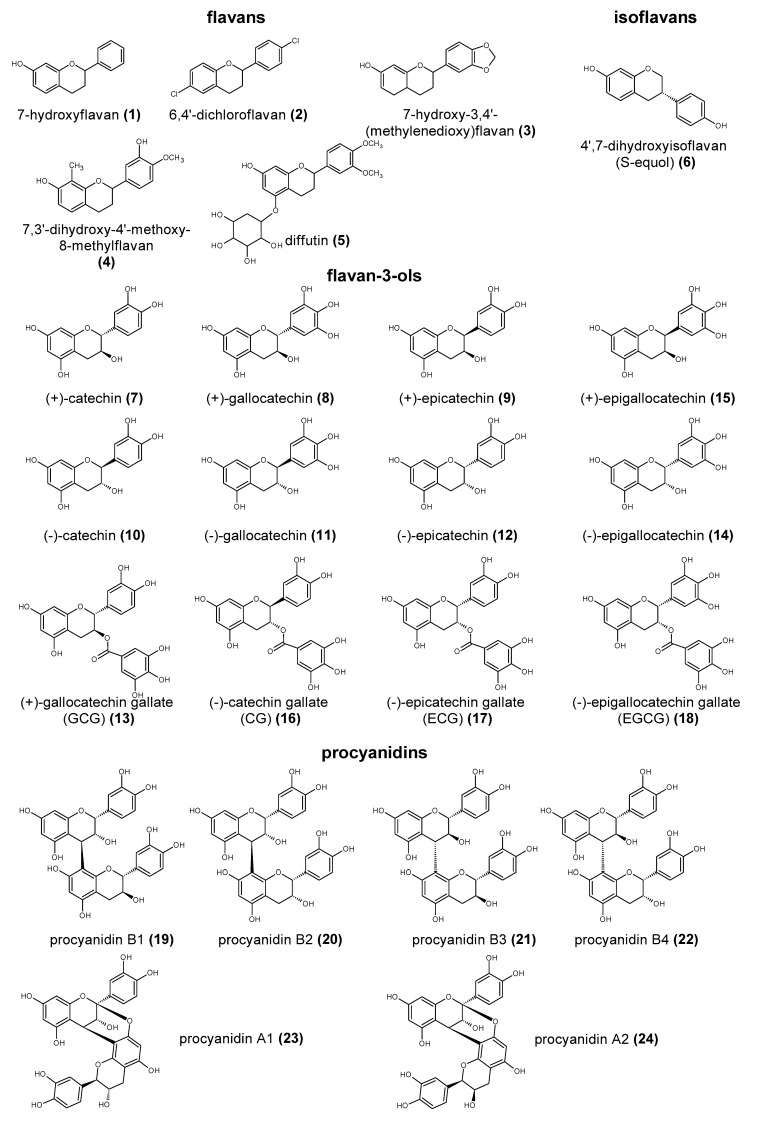
The chemical structure of various classes of flavonoids. Based on [16,24,33,34,35,36,37,38,39,40].

**Figure 2 antioxidants-10-00188-f002:**
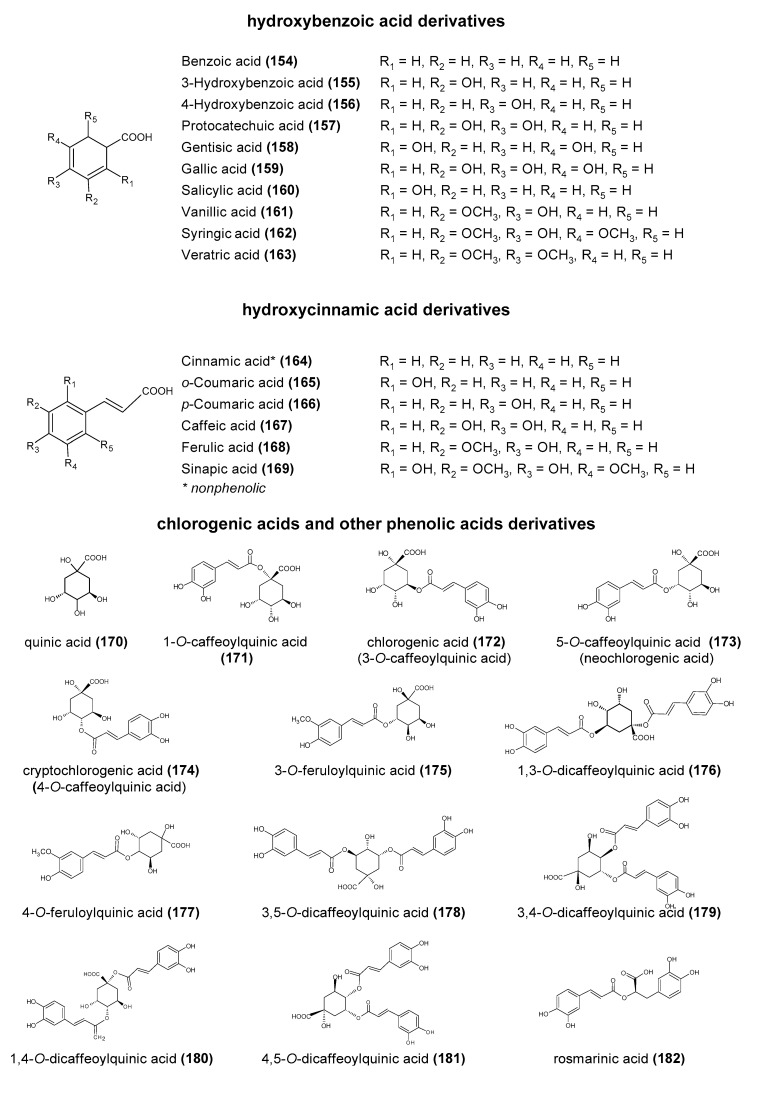
The chemical structure of various groups of non-flavonoid polyphenols. Based on [16,24,33,34,35,36,37,38,39,40].

**Figure 3 antioxidants-10-00188-f003:**
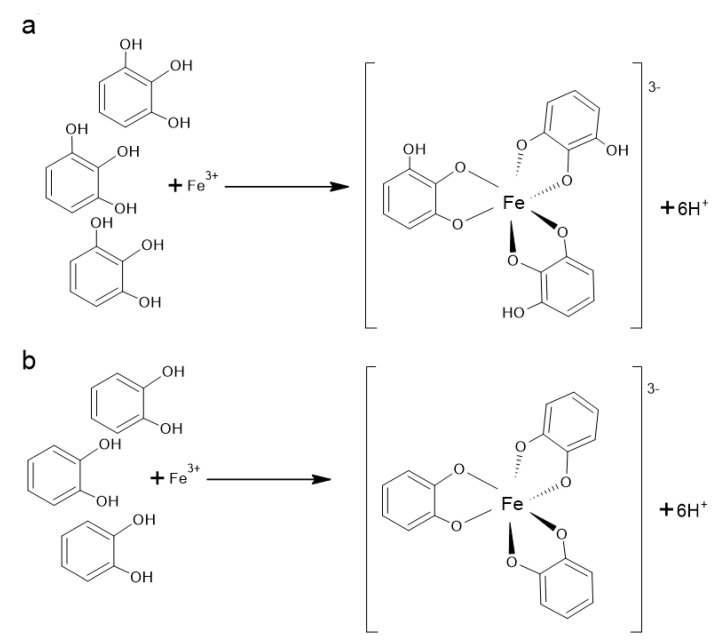
Expected octahedral coordination geometry of general iron-polyphenol complexes, (**a**) gallol, (**b**) catechols. Coordination requires deprotonation of the polyphenol ligands. Based on [201].

**Figure 4 antioxidants-10-00188-f004:**
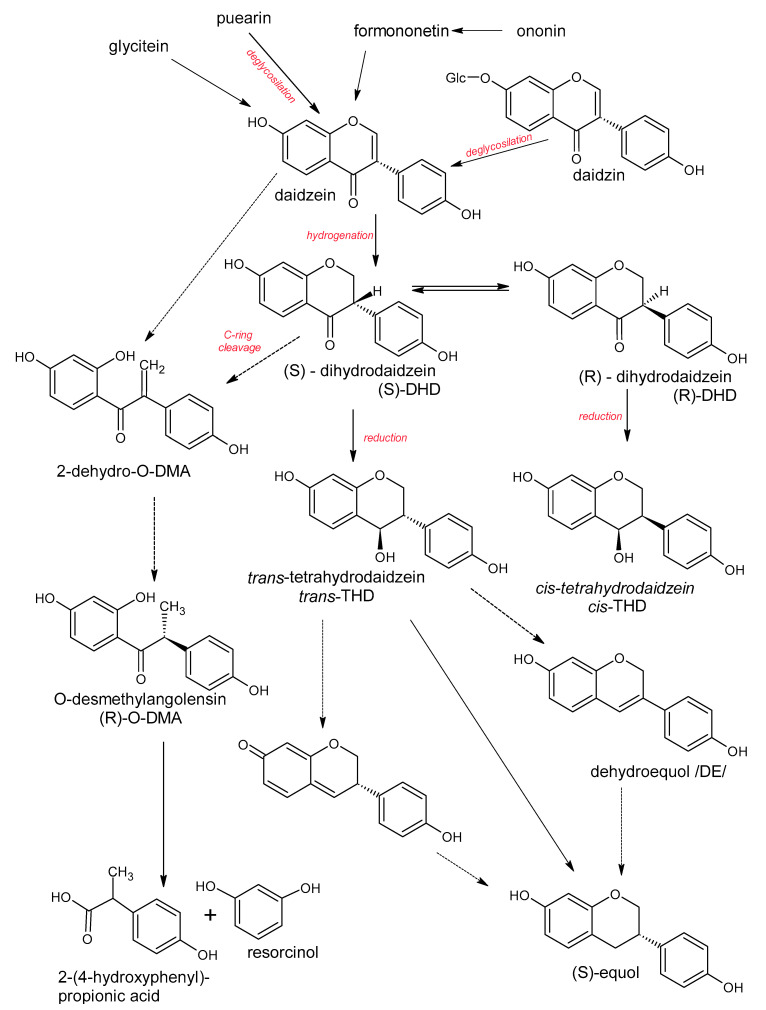
Possible pathways of microbial metabolism of daidzin and daidzein. Based on [236,237,238,239,240,241,242,243]. The dashed arrows indicate hypothesized reactions of microbiological degradation that were observed in vitro but were not reported in vivo.

**Figure 5 antioxidants-10-00188-f005:**
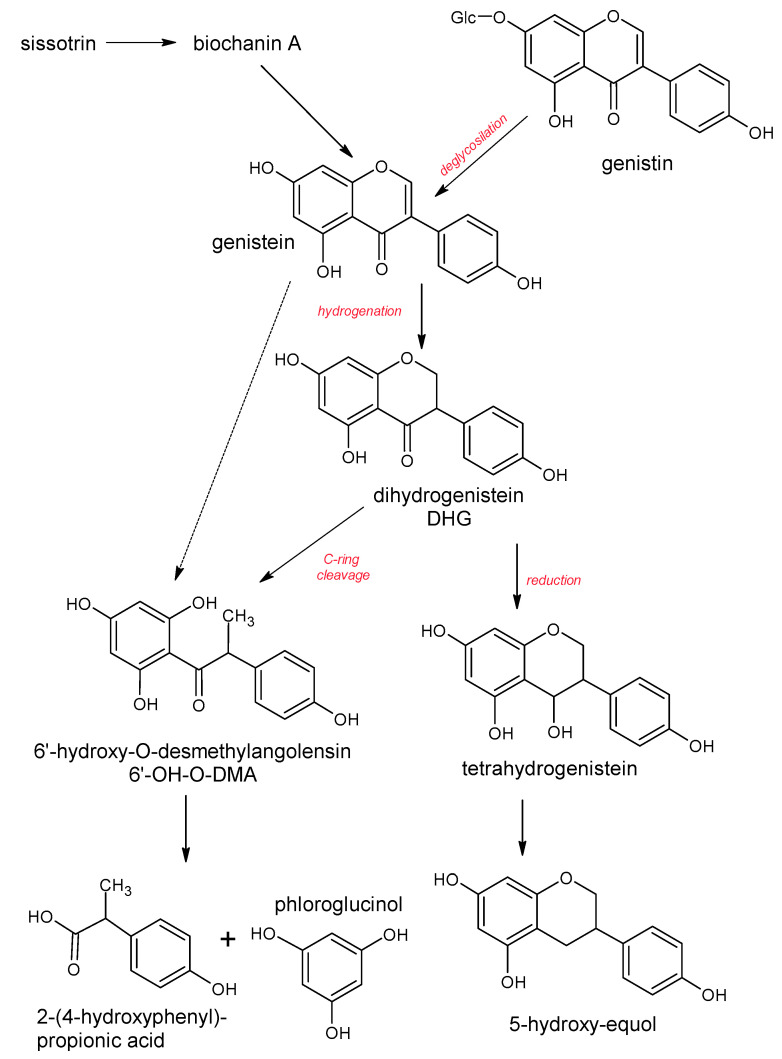
Possible pathways of microbial metabolism of genistin and genistein. Based on [236,237,238,239,240,241,242,243]. The dashed arrows indicate hypothesized pathways of microbiological degradation that were observed in vitro but were not reported in vivo.

**Figure 6 antioxidants-10-00188-f006:**
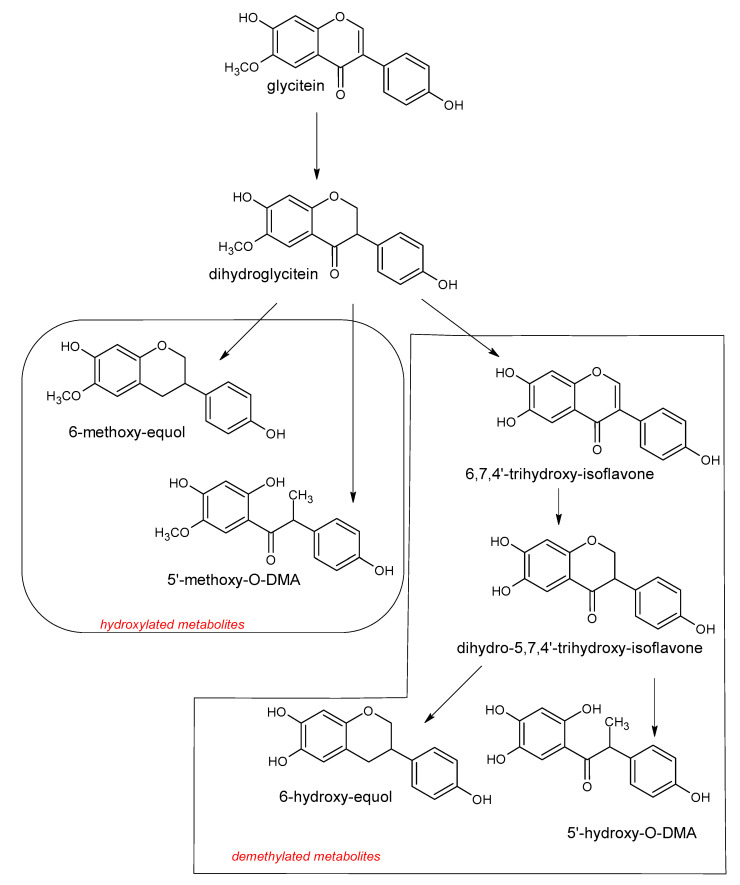
The pathways of bacterial metabolism of glycitin. Based on [237,242].

**Figure 7 antioxidants-10-00188-f007:**
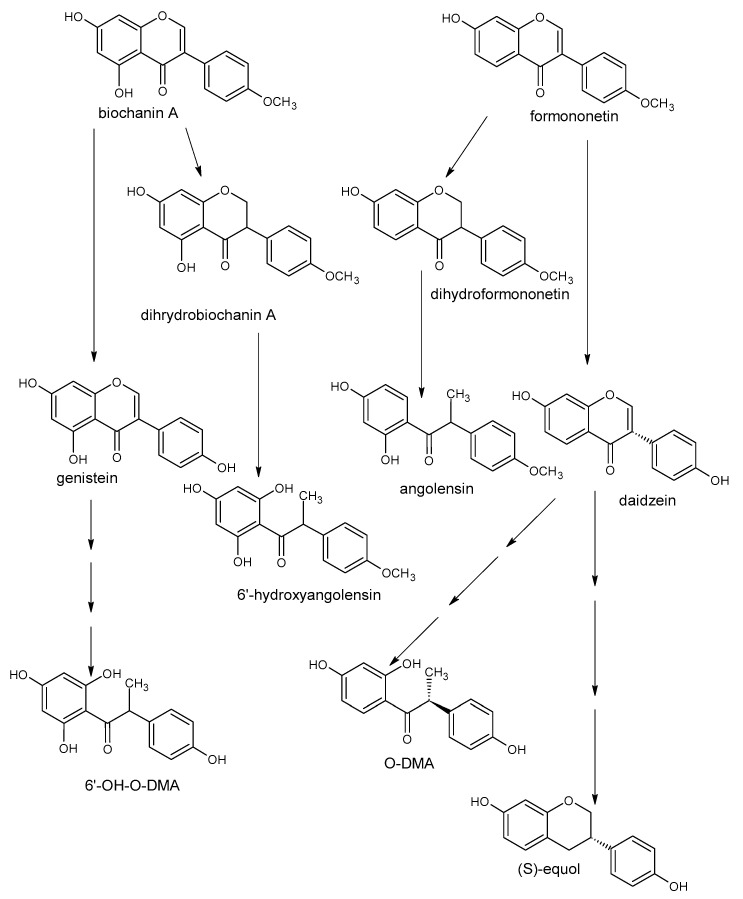
The human metabolism of formononetin and biochanin A. Based on [16,255,256].

**Figure 8 antioxidants-10-00188-f008:**
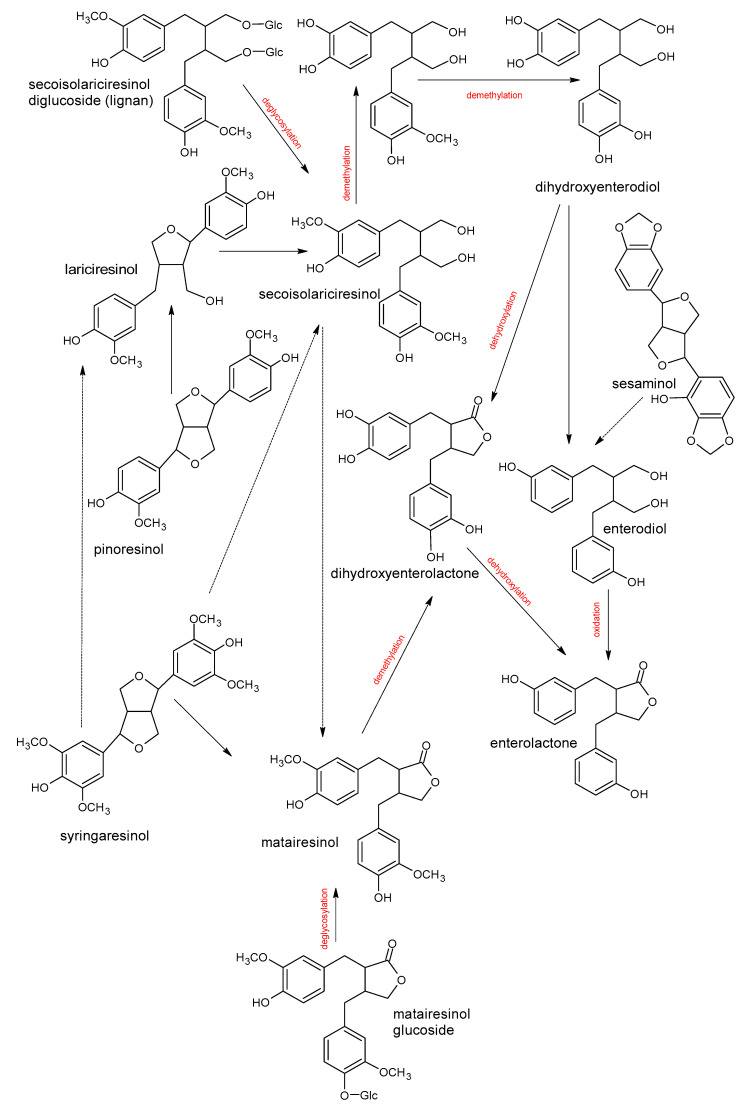
Lignans metabolism by gut microbiota. Based on [12,237,258,259,260,261]. The dashed arrows indicate hypothesized or multistep process.

**Figure 9 antioxidants-10-00188-f009:**
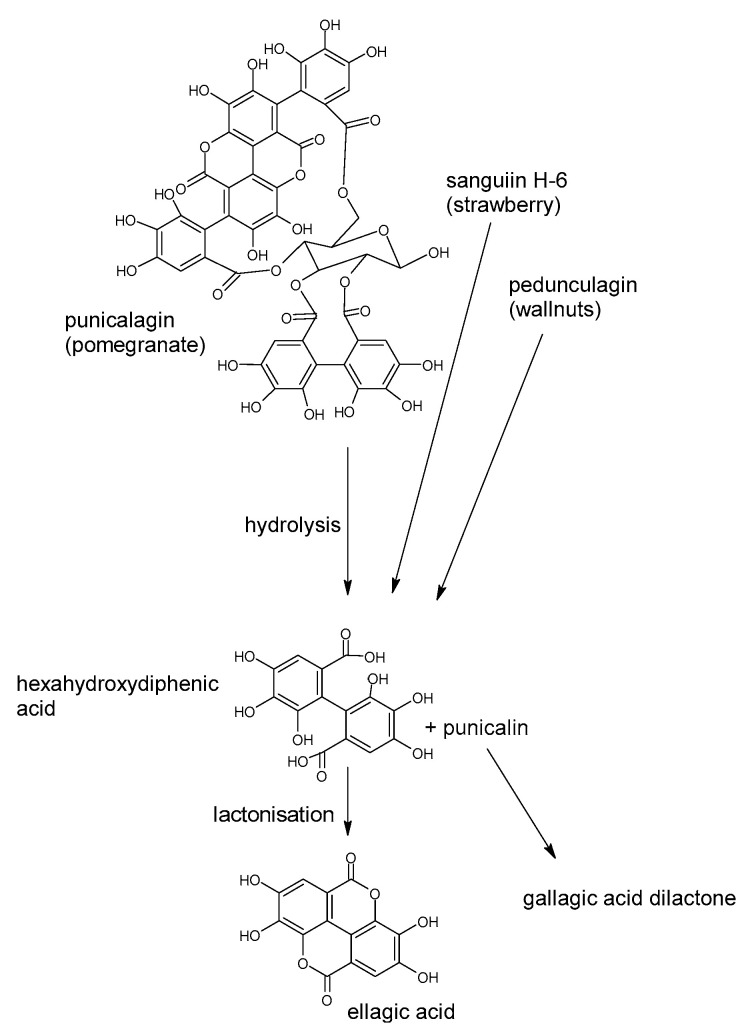
Degradation of ellagitannins to ellagic acid. Based on [268].

**Figure 10 antioxidants-10-00188-f010:**
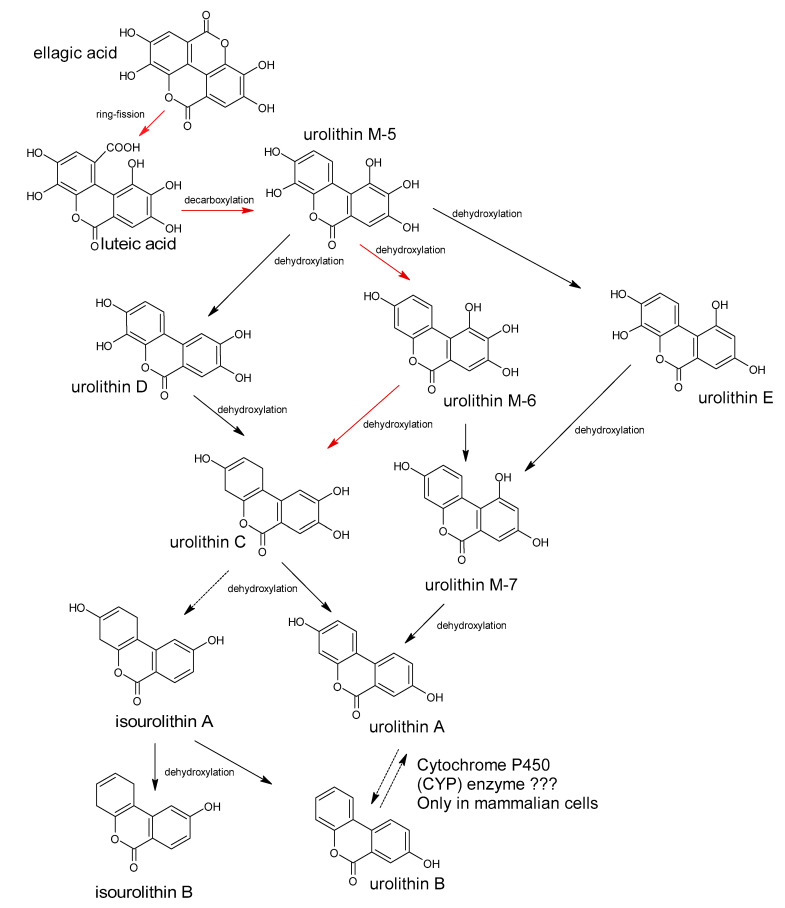
The bacterial metabolism of ellagic acid to urolithins and derivatives. The red arrows represent the pathways reported in *Gordonibacter urolithinfaciens* and *G. pamelaeae.* Based on [272,273,274].

**Figure 11 antioxidants-10-00188-f011:**
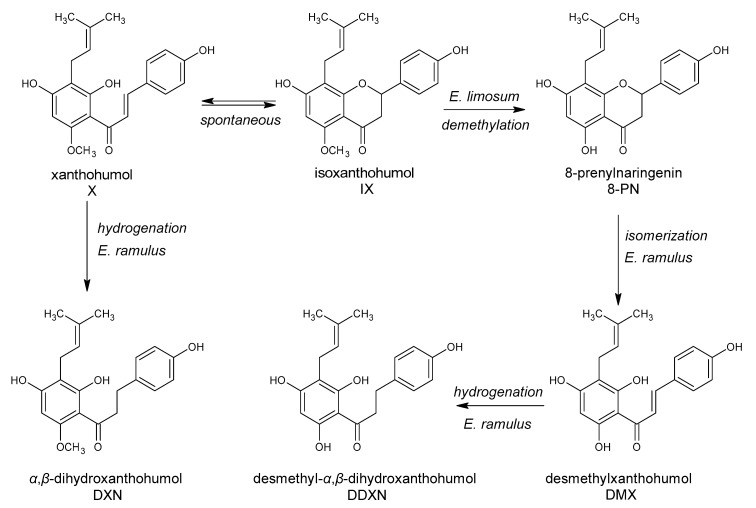
Bacterial biotransformation of prenylflavonoids xanthohumol and isoxanthohumol. Based on [275,276,279].

**Figure 12 antioxidants-10-00188-f012:**
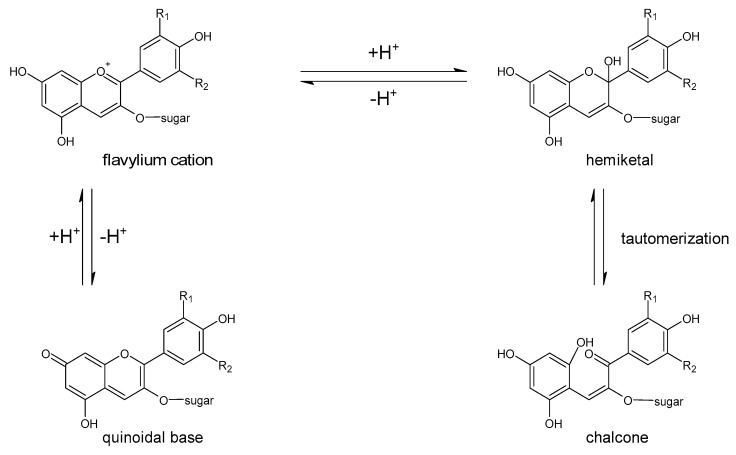
Anthocyanins equilibria [39].

**Figure 13 antioxidants-10-00188-f013:**
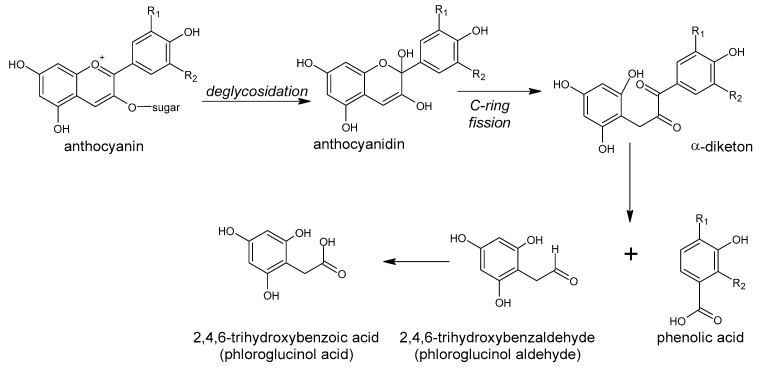
Main steps of anthocyanin degradation. Based on [274,282].

**Figure 14 antioxidants-10-00188-f014:**
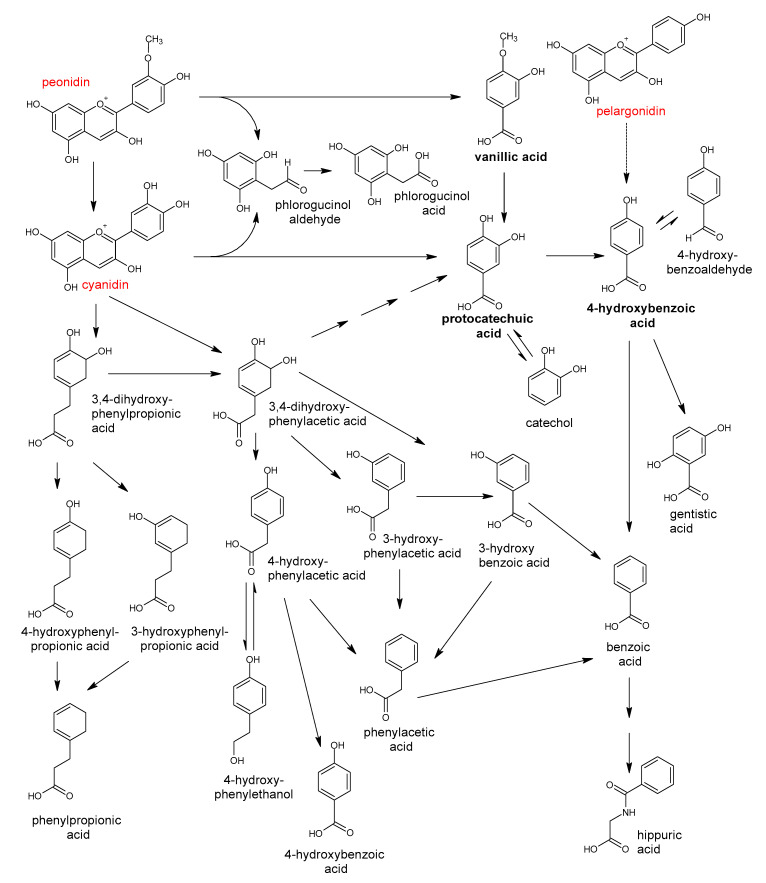
Biodegradation of anthocyanins and their main metabolites: cyanidin → 3,4-dihydroxybenzoic acid (protocatechuic acid), peonidin → 3-methoxy-4-hydroxybenzoic acid (vanillic acid) and pelargonidin → 4-hydroxybenzoic acid. Based on [282,284,287,288].

**Figure 15 antioxidants-10-00188-f015:**
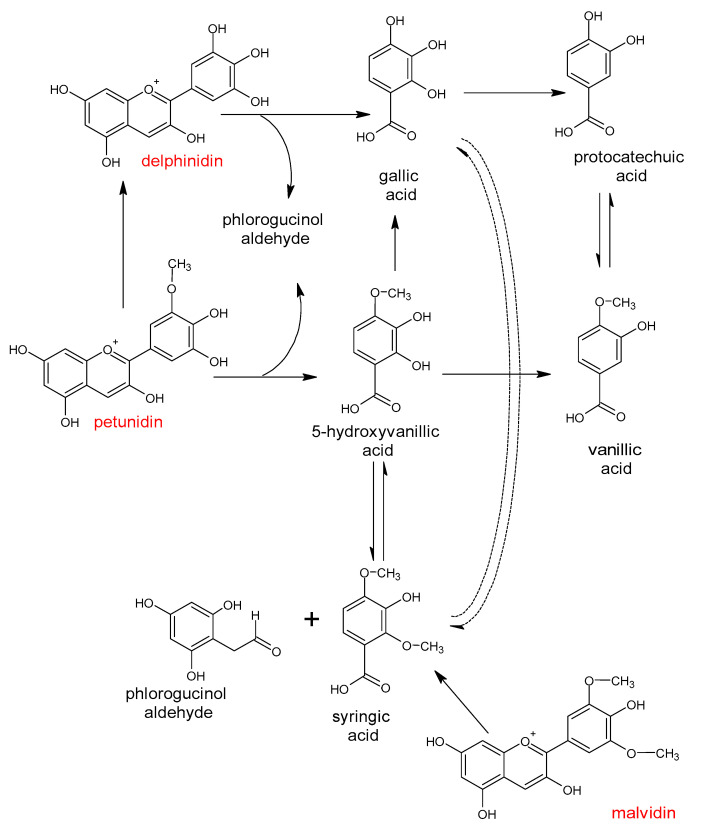
Biodegradation of malvidin, delphinidin and petunidin by intestinal bacteria. Based on [285,290,292].

**Figure 16 antioxidants-10-00188-f016:**
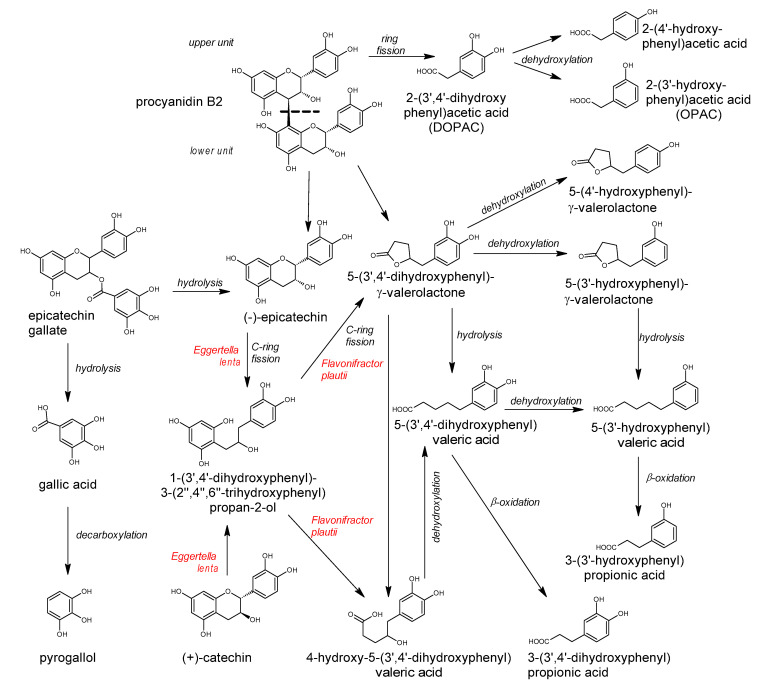
Microbial metabolism of procyanidin B2, (−)-epicatechin and (+)-catechin. Based on [274,291,294,295,296,297].

**Figure 17 antioxidants-10-00188-f017:**
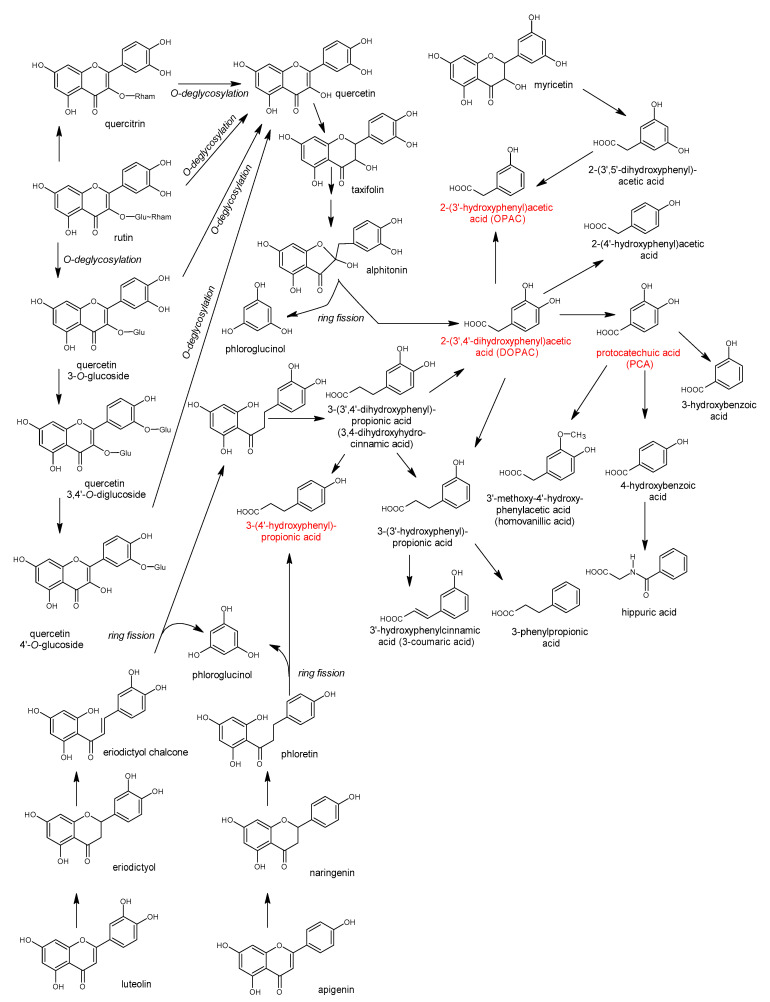
Possible pathways of the transformation of flavones and flavonols due to metabolism by intestinal bacteria. Based on [239,301,304,309,310].

**Figure 18 antioxidants-10-00188-f018:**
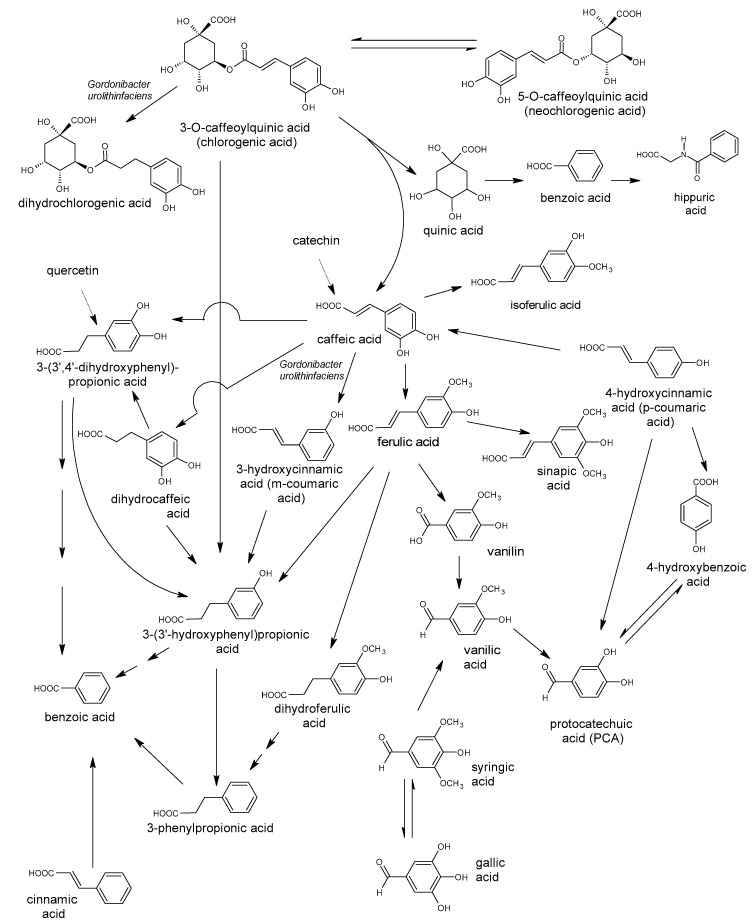
Pathways of some phenolic acid metabolism conducted by various bacteria. Based on [37,304,312,313,314,315].

**Figure 19 antioxidants-10-00188-f019:**
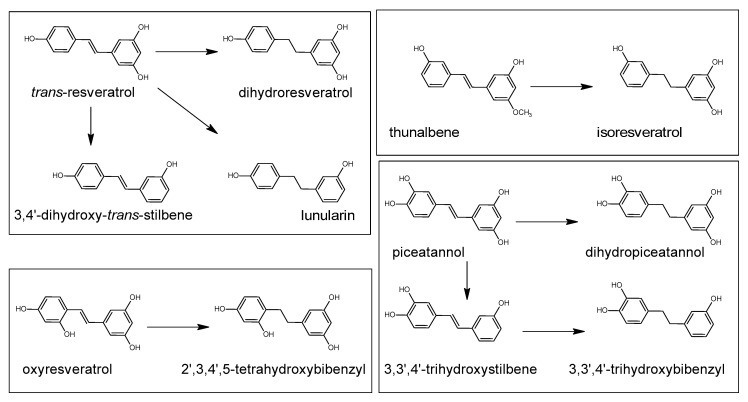
The effect of bacterial metabolism on resveratrol, oxyresveratrol, thunalbene and piceatannol. Based on [320,324].

**Figure 20 antioxidants-10-00188-f020:**
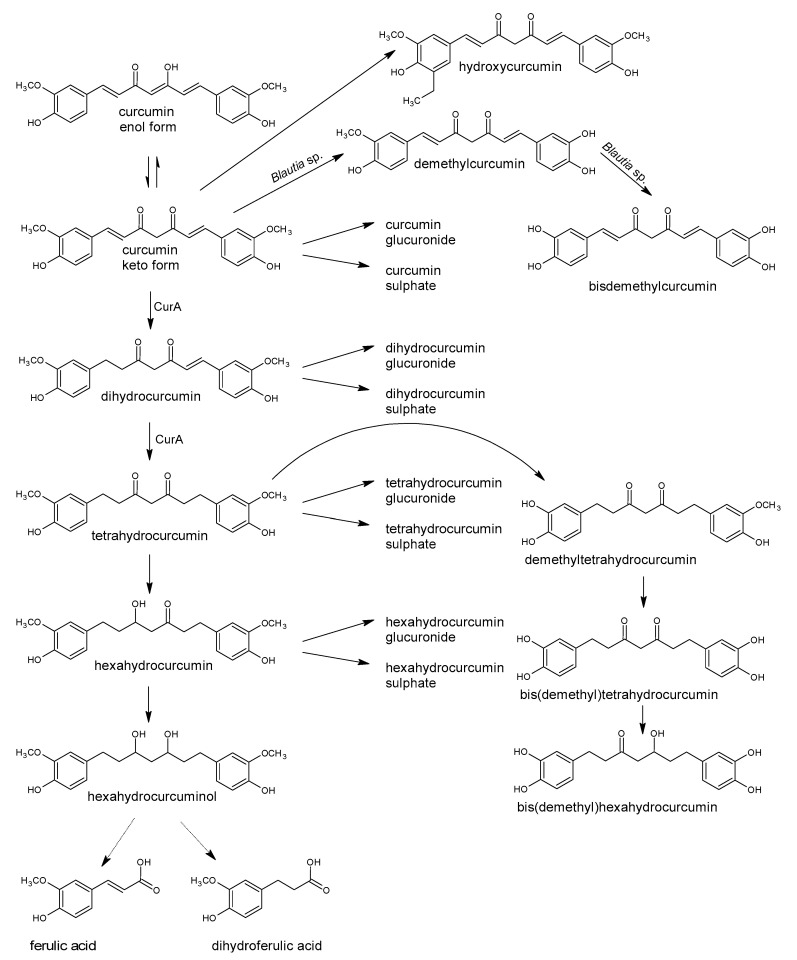
Metabolic pathway of curcumin. Reactions conducted by *E. coli* CurA and *Blautia* sp. are indicated. Based on [274,325,326,328,329,330].

**Table 1 antioxidants-10-00188-t001:** Examples of the negative and positive impact of pure polyphenols or plant preparations (aqueous, ethanolic, methanolic or other extracts, essential oils, enriched extracts) on food-associated bacteria or the representatives of the human gastrointestinal tract microbiota.

Plant Material, Preparation or Source	Pure Bioactive Compounds or Polyphenols Identified in Plant Material	Impact on Microorganisms	Reference
Ethanolic extracts of: rosemary, pomegranate peel, grape seed		Antimicrobial activity of rosemary extract against *S. aureus*, *E. coli*, *P. aeruginosa*, *K. pneumonia*, *B. subtilis*, *M. luteus* and *C. albicans*. Pomegranate peel extract inhibited all mentioned microorganism except *P. aeruginosa*. Inhibitory impact on *S. aureus*, *E. coli*, *B. subtilis*, *M. luteus* and *C. albicans*.	[41]
Ethanolic extract of sage (*Salvia officinalis*) flower, leaf and stem and essential oils	Manool, α-pinene, camphene, camphor, limonene, bornyl, 1,8-cineole, linalool, cis- and trans-thujone, acetate, α-humulene	Antibacterial activity against *B. subtilis*, *S. aureus*, *S. enteritidis*, *P. aeruginosa*, *E. coli*. Extracts had stronger antibacterial activity that the essential oils.	[42]
Ethanolic extracts of Heliotropum europaeum		Antimicrobial activity against *B. subtilis*, *E. coli*, *P. aeruginosa*, *P. fluorescens*, *S. aureus*, *MRSA*, *S. epidermidis*, *S. odorifera P. vulgaris*, *K. pneumoniae*, *E. faecium* and clinic isolate of *E. faecalis*. No impact on *E. aerogenes*, multidrug resistant strains of *E. coli* and *K. pneumoniae*, *E. faecalis* ATCC 29212, *L. monocytogenes*, *S. haemolyticus*, *S. boydi*, *A. baumannii*, and *S. flexneri*.	[43]
Ethanolic extracts from thyme (*Thymus vulgaris*) and sage (*Salvia officinalis*) leaves	Flavonoids tannins rosmarinic acid **(182)**, caffeic acid **(167)**, chlorogenic acid **(172)**, carnosol	Inhibited growth of *S. aureus, Vibrio tubiashii, L. pneumophila, M. luteus, Streptococcus* sp., *B. cereus.*	[44]
Aqueous and methanol extracts of *Salvadora persica* L.		Aqueous extract inhibited in dose-dependent manner all tested microorganisms, especially *Streptococcus mutans, S. faecalis, S. pyogenes*, as well as *S. aureus, L. acidophilus, P. aeruginosa*, and *C. albicans*. Methanol extract was inactive against *L*. *acidophilus* and *P. aeruginosa.*	[45]
Hydrodistilled essential oils from *Mentha piperita* and *Rosmarinus officinalis*	Mint: α-terpinene, piperitenone oxide, isomenthone, trans-carveol and β-caryophyllene. Rosemary: piperitone, α-pinene, linalool, camphor, 1,8-cineole	Inhibition of the growth of *Streptococcus mutans* and *S. pyogenes.* Inhibition of biofilm formation by *S. mutans*.	[46]
Phenolic extracts from blueberry, lingonberry, blackcurrant, raspberry, cloudberry, cranberry and strawberry Pure polyphenols	Extracts contained anthocyanins, flavonols, flavan-3-ols, hydroxycinnamate. myricetin **(114)**, luteolin **(49)**, apigenin **(38)**, kaempferol **(202)**, quercetin **(111)** rutin **(121)** isoquercitrin **(126)** (+)-catechin **(7)** phenolic acids: trans-cinnamic acid **(164)**, *m*-coumaric acid, caffeic acid **(167)**, ferulic acid **(168)**, chlorogenic acid **(172)**	Strong antimicrobial activity against *Salmonella enterica* ser. Thyphimurium SH5014, *E. coli* CM871 and *E. coli* 50. Inhibition of *Lactobacillus rhamnosus* and *L. rhamnosus* GG growth by cloudberry, raspberry and strawberry extracts at higher concentrations. Strawberry extract was effective inhibitor against *E. faecalis* and *Bifidobacterium lactis*. Strong inhibition of the growth of all lactic acid bacteria derived from the human gut, and *E. coli* growth by myricetin; but no impact on *Salmonella* Typhimurium and *Lactobacillus plantarum* from beer. Bacteriostatic impact of luteolin on some *Lactobacillus* species, *Bifidobacterium lactis* and *Enterococcus faecalis*, no impact on Gram-negative bacteria. No impact of (+)-catechin, kaempferol, apigenin, isoquercitrin, and rutin. Phenolic acids at concentration 0.5 mg/well phenolic acids inhibited the growth of *E. coli*, and *S. enterica*.	[47]
Methanolic extracts of six species of Hypericum	tannin, flavonoid and phenolic acids, among them quercitrin **(123)**, hyperoside **(120)**, isoquercitrin **(126)**, chlorogenic acid **(172)** were identified	*H. caprifoliatum* – the most active, inhibition of *S. aureus.* *H. polyanthemum* and *H. ternum* extracts antibacterial activity against *B. subtilis.* No activity of extracts against activity against *S. epidermidis* and *E. coli*.	[48]
Extracts of meadowsweet, willow herb, cloudberry, raspberry, bilberry and crowberry Extracts of white birch, pine, potato	Extract contained flavonoids and phenolic acid Pure compounds: quercetin **(111)**, morin **(110)**, rutin **(121)**, naringenin **(81)**, naringin **(85)**, kaempferol **(202)**	*E. coli* and *S. aureus* growth inhibition. Only *S. aureus* growth inhibition. Quercetin, morin, naringenin inhibited *S. aureus, S. epidermidis, E. coli, B. subtilis, M. luteous, P. aeruginosa*. Glycosides rutin and naringin were inactive. Kaempferol inhibited only *S. aureus*.	[49]
Peppermint essential oil and various extracts		Strong inhibition of *S. aureus* and *S. pyogenes* growth, less impact on of *E. coli* and *Klebsiella pneumoniae*. Petroleum ether, chloroform and ethyl acetate extracts were more effective antibacterial agents than ethanol and aqueous extracts. The strength of inhibition by extracts: *S. aureus > K. pneumoniae > S. pyogenes > E. coli*.	[50]
Peppermint oil Green tea polyphenols (GTP)	53 constituents of oil, e.g., menthol, menthone, neomenthol, menthofuran, (+)-limonene, piperiton, 3-octanol, *cis-*jasmone, mint lactone, (−)-myrtenol, piperitol, eugenol **(185)**, carvacrol **(183)**, 2-ethylfuran, ocimene (−)-epigallocatechin **(14)**, (−)-epigallocatechin-3-gallate **(18)**	Mentioned compounds had strong antibacterial activity against non-pathogenic *E. coli*. Oil, menthol, menthone and neomenthol killed the enterohemorrhagic strain *E. coli* O157:H7 at concentrations 400 µg/mL within 1 h. GTP inhibited *E. coli* O157:O7 growth at concentration of 800 µg/mL within 18 h. The synergistic effect was reported for peppermint oil + GTP and menthol + GTP.	[51]
Cranberry, blueberry and grape seed extracts and their synergy mixtures		Dose-dependent inhibitory activity against *Helicobacter pylori*. The synergy mixtures with higher concentrations of polymeric phenolics (procyanidins and tannins) had higher antimicrobial activity against *H. pylori*.	[52]
Chinese green tea extract	(−)-epicatechin gallate (ECG) **(17)**, (−)-epigallocatechin-3-gallate (EGCG) **(18)**	Crude green tea extract caused growth inhibition (by 44–100%) of food-borne pathogens *Escherichia coli* O157:H7, *Salmonella* Typhimurium DT104, *L. monocytogenes, S. aureus*, a diarrhoea food-poisoning pathogen *Bacillus cereus.* The lowest MIC90 values had EGCG: against *S. aureus* MSSA was 58 µg/mL, while against MRSA 37 µg/mL. ECG had MIC 309 and 105 µg/mL, respectively.	[53,54,55,56,57,58,59]
Green tea, Chung tea, Black tea	(−)-epicatechin (EC) **(12)**, (−)-epigallocatechin (EGC) **(14)**, (−)-epigallocatechin-3-gallate **(EGCG)** (18), (−)-epicatechin gallate (ECG) **(17)**, Teaflavins, quercetin **(111)**	EC and EGC strongly inhibited *S. aureus* ATCC 29213 (MIC 12.5 µg/mL), while ECG had MIC 50 µg/mL, ECG and EGCG strongly inhibited *E. coli* ATCC 25922 (MIC 12.5 µg/mL). ECG inhibited *Enterobacter cloacae* 1321E (MIC 12.5 µg/mL) and *S. pyogenes* (MIC 25 µg/mL), EC inhibited the growth of *P. aeruginosa* (MIC 25 µg/mL)	[54]
9 Hot water extracts of various tea (oolong, green, black, white)	gallic acid **(159)**, quercetin **(111)**, caffeine, (+)-catechin **(7)**, (−)-epicatechin **(12)**, (−)-epigallocatechin **(14)**	5-min extracts inhibited the growth of Helicobacter pylori, but growth of probiotics Bifidobacterium longum and Lactobacillus acidophilus was not affected.	[55]
	(−)-epigallocatechin-3-gallate **(18)**	EGCG inhibited intracellular groth of Listeria monocytogenes in macrophages	[56]
	(−)-epigallocatechin-3-gallate **(18)**	EGCG inhibits major functions of cellular and surface proteins, leading to growth inhibition of Bacillus subtilis	[57]
	(−)-epigallocatechin-3-gallate **(18)**, (−)-epigallocatechin (EGC) **(14)**	EGCG inhibits *S. aureus*, MRSA, *S. mutans, E. coli, P. aeruginosa, K. pneumoniae.* EGC inhibits *E. coli*	[58]
	(+)-catechin **(7)**, (−)-epicatechin **(12)**	Catechin caused significant decrease in the growth of the *Clostridium histolyticum* group and a marked increase in the growth of the beneficial bacterial group of *C. coccoides–Eubacterium rectale, Lactobacillus* spp. and *Bifidobacterium* spp. Epicatechin caused a significant increase in the growth of the *Eubacterium rectale–C. coccoides.*	[59]
	Pure polyphenols, inter alia: baicalein **(45)**, quercetin **(111)**, myricetin **(114)**, naringenin **(81)**, naringin **(85)**, hesperetin **(83)**, hesperidin **(86)**, resveratrol **(202)**, gallic acid **(159)**	Aglycones quercetin, naringenin, hesperetin inhibited growth of tested bacteria, while their glycoside did not. The lowest MIC were reported for: baicalein, myricetin, hesperetin and kaempferol against *E. coli* O157; baicalein nad myricetin against *S. aureus;* and baicalein against *Salmonella* Typhimurium, *Enterobacter sakazakii* and *Vibrio parahemolyticus*.	[60]
Citrus fruit	hesperetin **(83)**, naringenin **(81)**, poncirin **(90)**, diosmetin **(50)**	Inhibition of the growth of *H. pylori*.	[61]
	rutin **(121)**, quercetin **(111)**	Inhibition of *L. monocytogenes* growth.	[62]
	(+)-catechin **(7)**, quercetin **(111)**, naringenin **(81)**, hesperetin **(83)**, rutin **(121)**, naringin **(85)**, hesperidin **(86)**	Aglycones naringenin and hesperetin, inhibited growth of almost all analysed bacteria (MIC ≥ 250 μg/mL). Catechin, and glycosides naringin, hesperidin, and rutin had no impact on tested intestinal bacteria. Quercetin had strong inhibitory impact (MIC 20–50 μg/mL) on *Ruminococcus gauvreauii, Bacteroides galacturonicus* and *Lactobacillus* sp. growth.	[63]
	13 phenolic acids: benzoic acid **(154), ** 3-hydroxybenzoic acid **(155)**, 4-hydroxybenzoic acid **(156)**, 4-hydroxy-3-methoxy-benzoic acid, 3,4-dihydroxybenzoic acid, phenylpropionic acid **(188)**, 3-hydroxyphenylpropionic acid **(187)**, 4-hydroxyphenylpropionic acid **(189)**, 3,4-dihydroxyphenylpro-pionic acids, phenylacetic acid **(186)**, 3-hydroxyphenylacetic acid, 4-hydroxyphenylacetic acid, 3,4-dihydroxyphenylacetic acid	No impact on *P. aeruginosa* PAO1. Impact on *E. coli* was strain dependent. *E. coli* ATCC 25922 inhibition by benzoic and 4-hydroxy-3-methoxy-benzoic, phenylacetic and phenylpropionic acids at a concentration of 1000 mg/mL. *E. coli* O157:H7 (CECT 5947) was susceptible to benzoic, 3-hydroxybenzoic, 4-hydroxybenzoic, 4-hydroxy-3-methoxy-benzoic acids, phenylacetic, 3-hydroxy-phenylacetic, 4-hydroxyphenylacetic and 3,4-dihydroxyphenylacetic acids, phenylpropionic, 3-hydroxyphenylpropionic acid and 4-hydroxyphenylpropionic acids. *Lactobacillus paraplantarum* LCH7 was the most susceptible to the action of phenolic acids, while *L. fermentum* LPH1 was the most resistant. The susceptibility was strain-dependent. The most active compound were: 4-hydroxybenzoic acid for *L. fermentum* CECT 5716, *L. fermentum* LPH1, *L. brevis* LCH23, and *L. plantarum* LCH17; 4-hydroxybenzoic acid.and phenylpropionic acid for *L. paraplantarum* LCH7 and *L. coryniformis* CECT 5711, while 3-hydroxyphenylpropionic acid for *L. fermentum* CECT 5716.	[64]
Extracts from 3 *Eucalyptus* species	2’,6’-dihydroxy-3’-methyl-4’-methoxy-dihydrochalcone, eucalyptin **(48)**, 8-desmethyl-eucalyptin	Significantly inhibited growth of Gram-positive bacteria: *S. aureus, MRSA, B. cereus, E. faecalis, Alicyclobacillus acidoterrestris, Propionibacterium acnes.* Not show strong antibacterial activity against Gram-negative *E. coli* and *P. putida.*	[65]
Essential oils from *Origanum vulgare* and *Thymus vulgaris* *Ocinum basilicum* oil		Minimal Bactericidal Concentration /MBC/ ≤ 5 mg/mL against pathogens (*E. coli, Salmonella Enteritidis*, and *S.* Typhimurium), and beneficial bacteria (*Lactobacillus acidophilus* and *Bifidobacterium breve*). Higher activity against pathogenic bacteria (MBCs ≤ 10 mg/mL) than beneficial bacteria (MBCs of 80 mg/mL).	[66]
	13 common flavonoids (flavones, flavonols, flavanones) and 6 organic acids (aliphatic and aromatic acids) kaempferol **(108)**, quercetin **(111)**, chlorogenic acid **(172)**, salicylic acid **(160)**	Antimicrobial activity against Gram-negative bacteria *E. coli* and *P. aeruginosa*, higher than impact on Gram-positive ones: *E. faecalis* and *S. aureus*. Kaempferol, quercetin and chlorogenic acid had no significant influence on *P. aeruginosa.* Salicylic acid – the highest inhibitory activity against all tested bacterial species (MIC = 250–500 μg/mL).	[67]
A flavan-3-ol enriched grape seed extract		Inhibited growth of *Streptococcus thermophilus, Bifidobacterium lactis* BB12, *Lactobacillus fermentum, L. acidophilus* and *L. vaginalis.* Stimulated growth of some *Lactobacillus plantarum, L. casei*, and *L. bulgaricus* strains. No impact on *Bifidobacterium breve* 26M2 and *B. bifidum* HDD541 growth	[68]
	naringenin **(85)**, hes-peridin **(86)**, rutin **(121)**, quercetin **(111)**, gal-lic acid **(159)**, caffeic acid **(167)**, *p*-coumaric acid **(166)**, ferulic acid **(168)**, chlorogenic acid **(172)**, vanillic acid **(161), ** sinapic acid **(169)**, hesperidin **(86)**, quercetin **(111)**	All polyphenols influenced the growth of *Bifidobacterium adolescentis* and *B. bifidum* was assessed. Both the stimulatory and inhibitory effects of polyphenols were observed. Coumaric acid had strongest stimulatory effect on *B. bifidum*, while vanillic and caffeic acid stimulated *B. adolescentis*. Inhibitory dose-dependent impact of hesperidin and quercetin, on *B. bifidum* and *B. adolescentis*.	[69]
Tea phenolics and their derivatives	3-phenylpropionic acid **(188)**, caffeic acid **(167)**, gallic acid **(159)**	Inhibition of pathogenic *Clostridium perfringens, C. difficile* and *Bacteroides* spp. Less effect on commensal *Clostridium* spp., *Bifidobacterium* spp. and probiotic *Lactobacillus* sp. The growth of *Lactobacillus* ssp. and *Bifidobacterium* spp. inhibited by caffeic acid, 3-phenylpropionic acid and to the lesser extent by gallic acid, while *Lactobacillus casei* Shirota growth inhibited only by 3-phenylpropionic acid.	[70]
Red wine polyphenols		Stimulated growth of *Bifidobacterium* and *Lactobacillus*.	[71]
Tannin-rich diet		Significantly decreased growth of *Clostridium* spp. with a corresponding shift toward Entero-bacteriaceae and Bacteroides	[72]
Pomegranate extract	ellagitannins	Enhancing the growth of beneficial bacteria *Akkermansia muciniphila.*	[73]
	resveratrol **(202)**	Enhancing the growth of beneficial bacteria *Akkermansia muciniphila, Lactobacillus* and *Bifidobacterium*.	[74] [75]
	polymeric procyanidins	Enhancing the growth of beneficial bacteria *Akkermansia muciniphila.* Markedly decreased the Firmicutes/Bacteroidetes ratio	[76]
Grape polyphenols		Enhancing the growth of beneficial bacteria *Akkermansia muciniphila*, and decreasing the proportion of Firmicutes to Bacteroidetes.	[77]
	rutin **(121)**	The beneficial impact on *Lactobacillus* and *Bifidobacterium*	[63]
Cocoa	polyphenols	The beneficial impact on *Lactobacillus* and *Bifidobacterium*	[78]
Red wine polyphenols		Significantly increased the number of Enterococcus, *Prevotella*, *Bacteroides*, *Bifidobacterium*, *Bacteroides uniformis*, *Eggerthella lenta*, and *Blautia* *coccoides*–*Eubacterium rectale* groups	[79]
Blueberry	anthocyanidins	The beneficial impact on *Lactobacillus acidophilus* and *Bifidobacterium* spp	[80]
Red wine	polyphenols	Significantly increase of the number of faecal bifidobacteria and *Lactobacillus* (intestinal barrier protectors), *Faecalibacterium prausnitzii* and *Roseburia* (butyrate-producing bacteria)	[81]
	flavan-3-ols epigallocatechin gallate **(18)**, (−)-epigallocatechin **(14)**, procyanidin B1 **(19)**, procyanidin B2 **(20)**	Significant inhibition of *L. acidophilus* LA-5 and *L. plantarum* IFPL379 adhesion, except 4 compounds. Enhanced *L. acidophilus* LA-5 adhesion to Caco-2 cells. Increased *L. casei* LC115 adhesion to Caco-2 cells. Increased the adhesion of *Lactobacillus casei* LC115 to HT-29 cells.	[82]
Extract from apples	procyanidin B2 **(20)**, chlorogenic acid **(172)**	The increase of adhesion of *Lactobacillus gasseri* and *Lactobacillus casei* to intestinal epithelial cells.	[83]
Cranberry	proanthocyanidins with A-type linkages	Inhibition of the adhesion of both antibiotic-susceptible and antibiotic-resistant strains of uropathogenic P-fimbriated *E. coli*.	[84,85]
	anthocyanidins: pelargonidin **(25)**, cyanidin **(26)**, delphinidin **(27)**, cyanidin-3-glucoside **(36)**	Inhibition of the growth of *E. coli* strain CM871. No effect on *E. faecalis* strain E-203, *S. enterica* SH-5014 and strains of the *Lactobacillus* and *Bifidobacterium*.	[47]
Wine extract	quercetin **(111)**, flavan-3-ols, anthocyanins	No impact on species belonging to the *Lactobacillus, Enterococcus, Bacteroides*, and *Bifidobacterium* genera.	[86]
Grape seeds and pomace	tannic acid **(215)**	Potent growth-promoting effects on *L. acidophilus.*	[87]
	gallic acid **(159)**, and free anthocyanins, vanillic acid **(161)**, protocatechuic acid **(157)**	Activated cell growth and the rate malolactic fermentation of *Leuconostoc oenos*. Vanillic acid showed a slight inhibiting effect, while protocatechuic acid had no effect.	[88]

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
