# Peer review of "The Interactions between Polyphenols and Microorganisms, Especially Gut Microbiota"

_antioxidants, 2021, doi:10.3390/antiox10020188_

Round 1
Reviewer 1 Report
Paper: The interactions between polyphenols and microorganisms, especially gut microbiota described by Małgorzata Makarewicz et al.
This review presents the comprehensive knowledge about the bi-directional relationship between polyphenols and the gut microbiome. This paper before accepting needs some correction:
- Key words- should be different than in title
- Introduction section should be making shorter by 70%, especially section about polyphenols, some Fig 1 present basic information presented in many other articles and books
- Section 2 –this paper is for polyphenols action not for essential oils therefor my proposition is detected this part. Additionally in next section only polyphenols as bioactive compounds was analysed.
- Additionally, mechanistic description some information as Cueva et al. [64] analysed the or Peppermint essential oil was or Dall’Agnol et al. [48] analysed or Duda-Chodak [63] demonstrated, etc. That will be better present tis information from section 2 as in Table. This will be much clearer for readers and present what kind of fraction is bioactive in some plants.
- Line 188- how is possible: The alcoholic extract and essential oils of Salvia officinalis (sage) contain high concentrations of phenolic acids and …. ??? should be Alcoholic extract of …. rich in ….
- Conclusion section – add some more information about mechanism and interaction between microbial and polyphenols and further marks
- What about microbiota of fermented food and drinks? How they influence on polyphenols? add some information as additionally section.
Author Response
Dear Reviewer
I would like to thank you very much for reading the article and for your suggestions that help to improve our review. Our resposnse to the Reviewer's comments are in blue color.
Paper: The interactions between polyphenols and microorganisms, especially gut microbiota described by Małgorzata Makarewicz et al.
This review presents the comprehensive knowledge about the bi-directional relationship between polyphenols and the gut microbiome. This paper before accepting needs some correction:
- Key words- should be different than in title
Response: The only term common for title and key words was “polyphenols”. Taking into account the Reviewer suggestion we deleted it from keywords. Moreover, we added 2 new key words: “inhibition” as polyphenols in many cases inhibit intestinal microbiota and “diversity”. We believe that both are relevant to the content of the manuscript.
- Introduction section should be making shorter by 70%, especially section about polyphenols, some Fig 1 present basic information presented in many other articles and books
Response: We can agree that the section about polyphenols should not necessarily be an introduction, but rather a separate chapter. However, we think the data it contains are important for the whole review. That part was excluded and now it constitutes the separate chapter 2, not introduction.
We are aware that some parts of Fig 1, and also Fig. 2, present basic information presented in many other articles and books. Nevertheless, that was the idea of this review - collect all in one place, therefore the reader don't need to browse through other sources.
- Section 2 – this paper is for polyphenols action not for essential oils therefor my proposition is detected this part. Additionally in next section only polyphenols as bioactive compounds was analysed.
Response: We’re not sure if we well understood the reviewer. Essentially, oils are complex mixtures of hundreds of aromatic compounds, among them also polyphenols can be present. Therefore, essential oils were mentioned in the paper as ingredients they contain, inter alia polyphenols, have significant antimicrobial properties. Moreover, oils can be obtained from various parts of plant using different extraction techniques, therefore, depending of the extraction method, solvent used and plant tissue, the profile of bioactive compounds in essential oil differs. Thus, they can contain more or less polyphenols components, but it should not be neglected or ignored.
- Additionally, mechanistic description some information as Cueva et al. [64] analysed the or Peppermint essential oil was or Dall’Agnol et al. [48] analysed or Duda-Chodak [63] demonstrated, etc. That will be better present tis information from section 2 as in Table. This will be much clearer for readers and present what kind of fraction is bioactive in some plants.
Thank you very much for this suggestion. While writing the manuscript, we considered this (tabular) version of the data presentation, but we were a bit apprehensive. As we can see, wrongly. Therefore, the section 2 (formerly), now 3, was rewritten and majority of data it contained are now presented in Table 1. In some cases we could also add more data, especially about bioactive compounds present in plant material.
We hope, that the information it presents is now clearer and easier to read.
- Line 188- how is possible: The alcoholic extract and essential oils of Salvia officinalis (sage) contain high concentrations of phenolic acids and …. ??? should be Alcoholic extract of …. rich in ….
Response: Of course it can be rich. Majority of polyphenols is poorly soluble in water, so addition of alcohol enhances the efficiency of polyphenols extraction from plant tissues. Alcoholic extracts often contain more polyphenols than aqueous extracts. Especially when alcohol is used in concentrations between 30-70%, but it depends both on alcohol kind (methanol, ethanol…) and plant raw material (plant species, plant tissue or part: flower, leaves, root etc.) as well as the extraction method (maceration, infusions, with/without ultrasounds and many more). For example: https://doi.org/10.1016/j.cbi.2007.01.020
https://doi.org/10.1016/j.indcrop.2012.08.029
https://doi.org/10.17660/ActaHortic.2004.629.16
- Conclusion section – add some more information about mechanism and interaction between microbial and polyphenols and further marks
We have added to the Conclusion section some summarizing information about the mechanisms and interaction between polyphenols and microorganism which were described in details in the manuscript (lines 1866-1878).
- What about microbiota of fermented food and drinks? How they influence on polyphenols? add some information as additionally section.
Response: We now that this could be interesting, but actually this subjects were not within the scope of our review. Moreover, present version is very long and providing new section with thoroughly compiled new data would make the article too long. However, fermented food contain mainly lactic acid bacteria (milk and not-milk fermented food and beverages, such as pickled cucumbers and other vegetables, sauerkraut, yoghurt, buttermilk, sour milk, butter, cream, koumiss, and others) that are included and described in the manuscript. As for alcohol-fermented foods, yeasts are the main microorganisms there, but that is a topic for a separate work, quite long.
Reviewer 2 Report
The interactions between polyphenols and microorganisms, especially gut microbiota
This review showed the relationships between polyphenols and microorganisms. This review is potentially interesting, however, several crucial problems to be solved.
This study summarized polyphenols, this was very interesting and meaningful. However, the association between these polyphenols and microorganisms were difficult to understand for general readers.
This reviewer recommends that the relationship between these polyphenols and microorganisms be summarized in tables
Author Response
First off all, we would like to thank you very much for reading the article and for your suggestions that help to improve our review. Our response for the reviewer's comments are below in blue color.
Comments and Suggestions for Authors
The interactions between polyphenols and microorganisms, especially gut microbiota
This review showed the relationships between polyphenols and microorganisms. This review is potentially interesting, however, several crucial problems to be solved.
This study summarized polyphenols, this was very interesting and meaningful. However, the association between these polyphenols and microorganisms were difficult to understand for general readers.
This reviewer recommends that the relationship between these polyphenols and microorganisms be summarized in tables
Response: We are not sure what kind of “crucial problems” the Reviewer wanted to be solved. We understood that was the presentation of the relationship between polyphenols and microorganism in the form of table. Therefore, the impact of various polyphenols (both positive and negative) on beneficial and harmful microorganisms connected with human gastrointestinal tract are now presented in the Table 1.
However, we can’t imagine the presentation of data from other sections (4 and 5) in such tabular form, as in this way a lot of important information would be lost and the discussion of various results is impossible.
Round 2
Reviewer 2 Report
It's easier to read. No further comments.